# Assessing long-distance RNA sequence connectivity via RNA-templated DNA–DNA ligation

Christian K Roy[1,2], Sara Olson[3], Brenton R Graveley[3], Phillip D Zamore[1,2]*, Melissa J Moore[1,2]*

[1]RNA Therapeutics Institute, Howard Hughes Medical Institute, University of Massachusetts Medical School, Worcester, United States; [2]Department of Biochemistry and Molecular Pharmacology, University of Massachusetts Medical School, Worcester, United States; [3]Institute for Systems Genomics, Department of Genetics and Developmental Biology, University of Connecticut Health Center, Farmington, United States

**Abstract** Many RNAs, including pre-mRNAs and long non-coding RNAs, can be thousands of nucleotides long and undergo complex post-transcriptional processing. Multiple sites of alternative splicing within a single gene exponentially increase the number of possible spliced isoforms, with most human genes currently estimated to express at least ten. To understand the mechanisms underlying these complex isoform expression patterns, methods are needed that faithfully maintain long-range exon connectivity information in individual RNA molecules. In this study, we describe SeqZip, a methodology that uses RNA-templated DNA–DNA ligation to retain and compress connectivity between distant sequences within single RNA molecules. Using this assay, we test proposed coordination between distant sites of alternative exon utilization in mouse *Fn1*, and we characterize the extraordinary exon diversity of *Drosophila melanogaster Dscam1*.

*For correspondence: phillip. zamore@umassmed.edu (PDZ); melissa.moore@umassmed.edu (MJM)

## Introduction

One of the most important drivers of metazoan gene expression is the ability to produce multiple mRNA isoforms from a single gene. Around 58% of *Drosophila melanogaster* genes and >95% of human genes produce more than one transcript (*Pan et al., 2008*; *Wang et al., 2008*; *Brown et al., 2014*), with most human genes expressing 10 or more distinct isoforms (*Djebali et al., 2012*). Alternative promoter use, alternative splicing, and alternative polyadenylation all contribute to isoform diversity. In genes with multiple alternative transcription start and/or pre-mRNA processing sites, their combinatorial potential exponentially increases the number of possible products, with some human genes predicted to express >100 mRNA isoforms. In *D. melanogaster*, the number of isoforms observed per gene correlates with open reading frame length, suggesting that isoform complexity is a function of transcript length (*Brown et al., 2014*). The current record holder in this regard is *Dscam1*, in which four regions of mutually exclusive cassette exons combine to generate a remarkable 38,016 distinct >7000 nt mRNAs, each encoding a unique protein isoform (*Schmucker et al., 2000*).

In *Dscam1*, the four regions of mutually exclusive cassette exon splicing are separated by one to eight constitutive exons. This feature of multiple alternative splicing regions separated by constitutive exons is shared by more than a quarter of human genes (*Fededa et al., 2005*). In many cases, these regions are separated by >500 nts, the current limit for contiguous sequence output on most deep sequencing platforms. Further, high-throughput sequencing of RNA (RNA-Seq) generally requires its reverse transcription, with the processivity of available reverse transcriptases (RTs) limiting even single

**eLife digest** A flow chart can show how an outcome can be achieved from a particular start point by breaking down an activity into a list of possible steps. Often, a flow chart contains several alternative steps, not all of which are taken every time the flow chart is used. The same can be said of genes, which are biological instructions that often contain many options within their DNA sequences.

Proteins—which perform many roles in cells—are built following the instructions contained in genes. First, the DNA sequence of the gene is copied. This produces a molecule of ribonucleic acid (RNA), which is able to move around the cell to find the machinery that can use the genetic information to make a protein. Genes and their RNA copies contain instructions with more steps—called exons—than are necessary to make a working protein, so extra exons are removed ('spliced') from the RNA copies. Different combinations of exons can be removed, so splicing can make different versions of the RNA called isoforms. These allow a single gene to build many different proteins. In fruit flies, for example, the different exons of the gene *Dscam1* can be spliced into one of 38,016 unique RNA isoforms.

Current technology only allows researchers to deduce the sequence of RNA molecules by combining sequences recorded from short fragments of the molecule. However, before splicing, RNA molecules tend to be much longer than this, so this restricts our understanding of the RNA isoforms found in cells. Here, Roy et al. devised and tested a new method called SeqZip to solve this problem.

SeqZip uses short fragments of DNA called ligamers that can only stick to the sections of RNA that will remain after the molecule has been spliced. After splicing, the ligamers can be stuck together to make a DNA replica of the spliced RNA. The end product is at least 49 times shorter than the original RNA, so it is easier to sequence. In addition, the combinations of the ligamers in the DNA replica show which exons of a specific gene are kept and which ones are spliced out.

To test the method, Roy et al. studied a mouse gene that has six RNA isoforms. SeqZip reduced the length of the RNA by five times and made it possible to measure how frequently the different isoforms naturally arise. Roy et al. also used SeqZip to work out which isoforms of the *Dscam1* gene are used at different stages in the life of fruit fly larvae. SeqZip can provide insights into how complex organisms like flies, mice, and humans have evolved with relatively few—a little over 20,000—genes in their genomes.

molecule cDNA sequencing (e.g., Pacific Biosciences) to <2500 nt (*Sharon et al., 2013*). Thus, existing high-throughput technologies cannot readily retain connectivity information between very distant sequences within individual mRNA molecules. Instead, full-length transcripts must be inferred by piecing together multiple short overlapping reads (*Garber et al., 2011*; *Grabherr et al., 2011*; *Haas et al., 2013*; *Boley et al., 2014*). For widely separated regions of alternative exon use, this loss of connectivity significantly limits our abilities to catalog isoform abundance and understand the mechanisms underlying alternative isoform generation.

Here, we describe SeqZip, a method for profiling multiple distant (>1000 nt) sites of alternative splicing within individual RNA molecules. SeqZip uses sets of DNA oligonucleotides termed 'ligamers'. Each ~40–60 nt ligamer hybridizes to the 5′ and 3′ ends of a single alternatively spliced exon or the beginning and end of a large block of constitutively included exons, looping out the sequences in between. These loops can be hundreds to thousands of nucleotides long. Juxtaposed ligamers hybridized to single RNA molecules are then joined by enzymatic ligation with T4 RNA ligase 2 (Rnl2) (*Ho and Shuman, 2002*). The resultant DNA ligation products both capture the intramolecular connectivity among exons of interest and compress the sequence space necessary to identify those exons. Exon connectivity is subsequently decoded by assessing the sizes or sequences of the ligation products. Because SeqZip does not include an RT step and is therefore not subject to RT processivity and template-switching limitations, it can be used to assess intramolecular connectivity between regions separated by thousands of nucleotides. Further, relative ligation product abundance accurately reports spliced isoform abundance in the original sample. As a proof-of-principle, we here used SeqZip to test proposed connectivity relationships among alternatively spliced exons in mouse *Fibronectin* (*Fn1*) and to define the molecular diversity of fly *Dscam1*.

## Results

### A reverse transcription-free method to assess sequence connectivity

The general idea of SeqZip is schematized in *Figure 1*. This method requires efficient ligation of multiple DNA oligonucleotides (oligos) hybridized to an RNA template with little or no non-templated ligation. Although many ligases can join DNA or RNA oligos hybridized to a DNA template (*Bullard and Bowater, 2006*), when we initiated this study, only T4 DNA ligase was reported to join DNA fragments templated by RNA (*Nilsson et al., 2001*). While T4 DNA ligase is the basis of multiple RNA-templated DNA ligation methods (*Nilsson et al., 2001*; *Yeakley et al., 2002*; *Conze et al., 2010*; *Li et al., 2012*), it also catalyzes non-templated DNA ligation (*Kuhn and Frank-Kamenetskii, 2005*), which would reduce SeqZip fidelity.

To find a suitable ligase for SeqZip, we tested the ability of several other commercially available enzymes to ligate four or five 5′ $^{32}$P-radiolabeled 20-nt DNA oligos hybridized to adjacent positions on either DNA or RNA (*Figure 2A*). Although all DNA ligases tested could efficiently join multiple oligos hybridized to the DNA template (*Figure 2A*; *Bullard and Bowater, 2006*), only T4 DNA ligase and RNA ligase 2 (Rnl2) joined the DNA oligos when hybridized to the RNA template. Of the two, Rnl2 was more active for RNA-templated DNA ligation (data not shown) and produced <1/7 as much non-templated product as T4 DNA ligase (*Figure 2A*). Moreover, Rnl2 could not ligate DNA oligos hybridized to the DNA template, eliminating the possibility of contaminating genomic DNA confounding SeqZip (*Figure 2A*). We note that Chlorella virus DNA ligase was recently commercialized for the purpose of RNA-templated DNA–DNA ligation (SplintR ligase; NEB) (*Lohman et al., 2014*). We found, however, that SplintR ligase produces more non-templated DNA–DNA ligation events than Rnl2 (*Figure 2—figure supplement 1*). Also, while our paper was under review, another group reported RNA-templated DNA–DNA ligation by Rnl2 (*Larman et al., 2014*), further validating its use in SeqZip.

The SeqZip design requires efficient ligation of multiple DNA oligos (ligamers), some spanning loops in the RNA template (*Figure 1*). To test the ability of Rnl2 to ligate these species, we designed four different 26 nt ligamers to loop out various lengths of a 307 nt transcript (*Figure 2B*). Each 26 nt ligamer contained 10 nt of complementarity on either side of the loop, with a 6 nt spacer opposite the loop. The four ligamers—individually, pairwise, in threes, or as a complete set—were annealed to the template RNA and incubated with Rnl2. Ligation products were only observed when ligamers bound

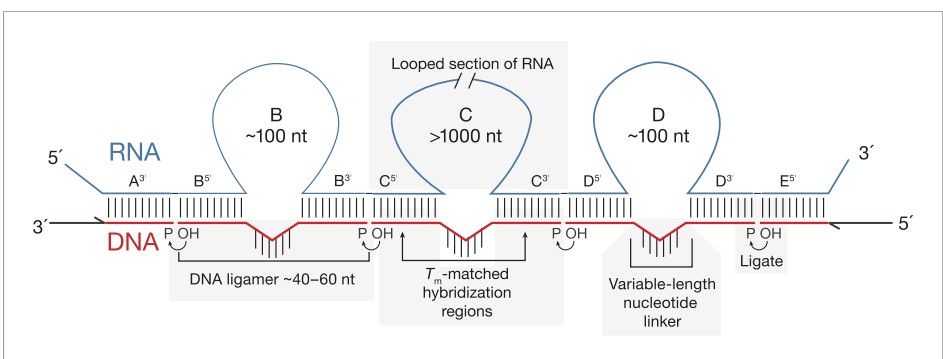

**Figure 1**. Principles of SeqZip. The target RNA is hybridized with a set of DNA oligonucleotides ('ligamers'). Ligamers targeting outermost sequences contain one region of complementarity and primer sequences for subsequent amplification. Internal ligamers contain two regions of complementarity separated by a spacer sequence. Hybridization of the internal ligamers causes the RNA between the hybridization sites to loop out. Hybridized ligamers are ligated, amplified, and analyzed.

The following figure supplements are available for figure 1:

**Figure supplement 1**. Ligamer design workflow.

**Figure supplement 2**. Other proposed uses of SeqZip.

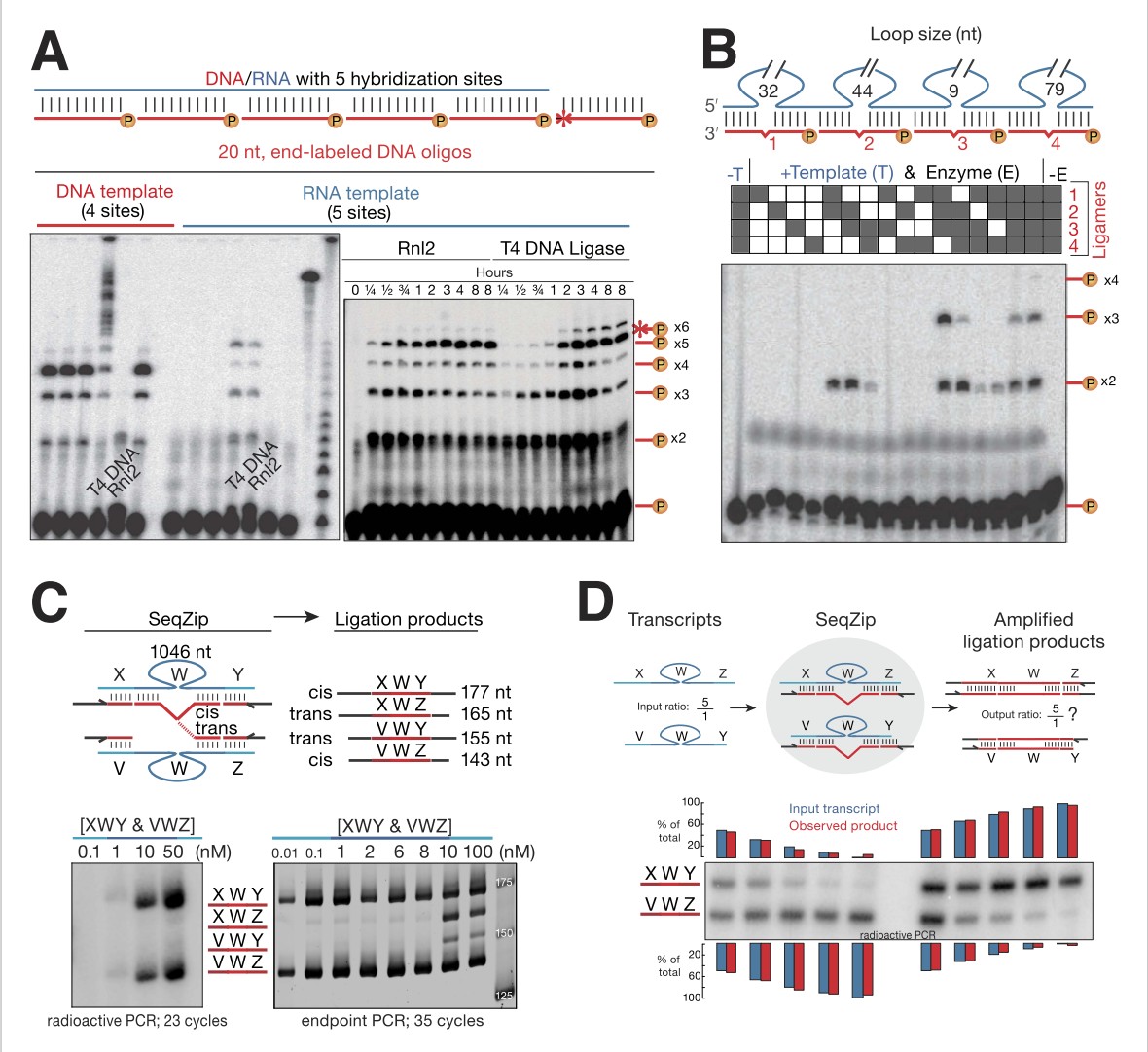

**Figure 2**. T4 RNA Ligase 2 catalyzes RNA-templated DNA-to-DNA ligation. (**A**) Left panel: ligase screen for RNA-templated DNA–DNA ligation activity. Ligases were incubated with an unlabeled single-stranded DNA (left) or RNA (right) template hybridized to a common pool of 5′ end $^{32}$P-labeled (circled P) DNA oligonucleotides for 1 hr. Both T4 DNA ligase and T4 RNA ligase 2 (Rnl2) catalyze RNA-templated DNA–DNA ligation. Also note the inability of Rnl2 to ligate >2 oligos on the DNA template. For both templates, ligases are left to right: *Tth* DNA ligase (Thermo), *Tsc* DNA ligase (Prokaria), Thermostable DNA ligase (Bioline), T4 DNA ligase (NEB), T4 Rnl2 (NEB), *E. coli* DNA ligase (NEB). The three rightmost lanes are $^{32}$P-oligos only, $^{32}$P-labeled RNA template, and a $^{32}$P-labeled low-molecular weight DNA ladder (NEB, N3233S). Right panel: Rnl2 and T4 DNA ligase time course for oligos hybridized to the RNA template. Templated ligation products (–x2 through –x5); non-templated ligation product (*–x6). (**B**) Rnl2 can join multiple $^{32}$P-labeled ligamers each looping out sections of the template but only when they are adjacently hybridized. Gray or white square: ligamer present or absent, respectively. No template (-T); no enzyme (-E). (**C**) *Cis*- and *trans*-transcript hybridization and ligation using a ligamer (W) spanning 1046 nt common to two RNAs (XWY and VWZ). Template concentrations (nM) were as indicated above each lane (ranging from 0.01 to 100 nM), ligamers were held constant at 10 nM. Left panel, phosphoimage; right panel, SybrGold stained. (**D**) The ability of SeqZip to accurately report on relative input RNA concentrations was investigated using various ratios of two RNAs (XWZ and VWY) and a six ligamer pool. Observed product ratios were calculated from radioactive PCR band intensities.

The following figure supplement is available for figure 2:

**Figure supplement 1**. Examination of SplintR ligase in the SeqZip assay.

to adjacent RNA sequences; four-way ligation products were obtained only when all ligamers were present. Thus, ligamers designed to loop out various lengths of a template RNA can be used to condense by more than twofold the information required to assess RNA connectivity—244 nt of the

target RNA was condensed to a 104 nt DNA. Subsequent ligamer designs condensed connectivity information by >49-fold.

## Minimal trans-transcript hybridization and ligation

A ligamer designed to loop out the sequences in between widely spaced regions of complementarity has the potential to bridge two RNA molecules. Such intermolecular (*trans*) hybridization would interfere with measurement of intramolecular (*cis*) RNA connectivity, producing artifacts akin to template switching in RT-based methods (*Figure 2C*; *Houseley and Tollervey, 2010*). To test the frequency of such *trans* events, we mixed two RNAs, each comprising a common 1106 nt internal sequence flanked by unique 5′ and 3′ sequences, with a ligamer set in which a single internal ligamer (W) looped out 1046 nt of the shared internal sequence (*Figure 2C*). Because the terminal ligamers (X, Y, V, and Z) varied in length, polymerase chain reaction (PCR) of SeqZip reactions yielded 177 and 143 nt *cis*-templated products and 165 and 155 nt *trans*-templated products. *Trans* hybridization of ligamer W, a tri-molecular interaction, should be much more sensitive to RNA concentration than bimolecular *cis* hybridization. Consistent with this, whereas *cis* products were detected by end point PCR at every target RNA concentration tested down to 0.01 nM, *trans* products were only detected when target RNAs were ≥10 nM, (*Figure 2C*, lower half). But, even when both targets were present at 50 nM, semi-quantitative radioactive PCR revealed that the *cis* hybridization products predominated (*Figure 2C*, lower left). Nonetheless, to disfavor *trans* hybridization, the general conditions for SeqZip described below use cellular RNA concentrations (10–40 ng/ml polyA+ RNA) at which most individual mRNAs are present at <1 nM.

To be useful as a quantitative method, SeqZip should accurately report on input RNA abundances. To test this, we mixed two target RNAs at ratios varying from 100:1 to 1:100 (a 100-fold dynamic range). Radioactive PCR revealed that their respective SeqZip product ratios paralleled these input ratios over the entire series (*Figure 2D*).

## SeqZip vs RT-based analysis of CD45 spliced isoforms

As a first test of SeqZip with a biological sample, we used it to assess alternative exon inclusion in endogenous human *CD45* (*PTPRC*) mRNA (*Zikherman and Weiss, 2008*). *CD45* isoforms contain various combinations of exons 4, 5, and 6 (*Figure 3A*). Jurkat cells (resembling naïve, primary T cells) predominantly express isoforms containing exons 5 and 6 (R56), only exon 5 (R5), or no cassette exon (R0). U-937 cells (resembling activated T cells) predominantly express the R56 isoform and one containing exons 4, 5, and 6 (R456; *Yeakley et al., 2002*). The three adjacent cassette exons occupy only 585 nt, making this region amenable to analysis by both reverse transcription and SeqZip. Reverse transcription-PCR (RT-PCR) products ranged from 365 to 848 nt, while SeqZip products ranged from 132 to 260 nt (*Figure 3B*), representing a ~threefold compression of connectivity information.

Using RT-PCR and SeqZip, we measured *CD45* isoforms from Jurkat or U-937 poly(A)-selected RNA or a 1:1 mixture of the two. Both methods reported the expected isoform abundances (*Figure 3B*). Importantly, even though SeqZip detection of R456, R56, R5, and R0 required different numbers of ligation events, all relative abundances were accurately reported, even in the mixture containing all four isoforms (*Figure 3B*, lower right).

## SeqZip and PacBio analysis of mouse *Fn1* isoform connectivity

For a more complex splicing pattern, we next turned to fibronectin (*Fn*). Mouse *Fn1* contains three well-characterized regions of alternative splicing: (1) the EDB exon included in embryos and adult brain but not other adult tissues, (2) the EDA exon variably included or excluded across multiple developmental and adult tissue types, and (3) the variable (V) region in which use of three alternative 3′ splice sites leads to inclusion of 120, 95, or 0 additional amino acids in the FN1 protein (*Figure 3C*). The original suggestion that an upstream splicing decision can affect a downstream splicing decision came from analysis of the EDA and V regions where it was reported that EDA exclusion promotes use of the promoter-proximal 3′ splice site ('120') in the V region (*Fededa et al., 2005*). The EDA and V regions are separated by six constitutively included exons, comprising 813 nt; thus, RT-PCR products including the EDA and V regions range from 1 to 1.6 kbp (*Figure 3D*). Both the overall length of the RT-PCR products and the extensive region of similar sequence identity in the middle that can promote template switching (see below) confound RT-PCR analysis of the six possible EDA and V exon combinations. In comparison, our SeqZip ligation products were >fivefold smaller (139–318 nt;

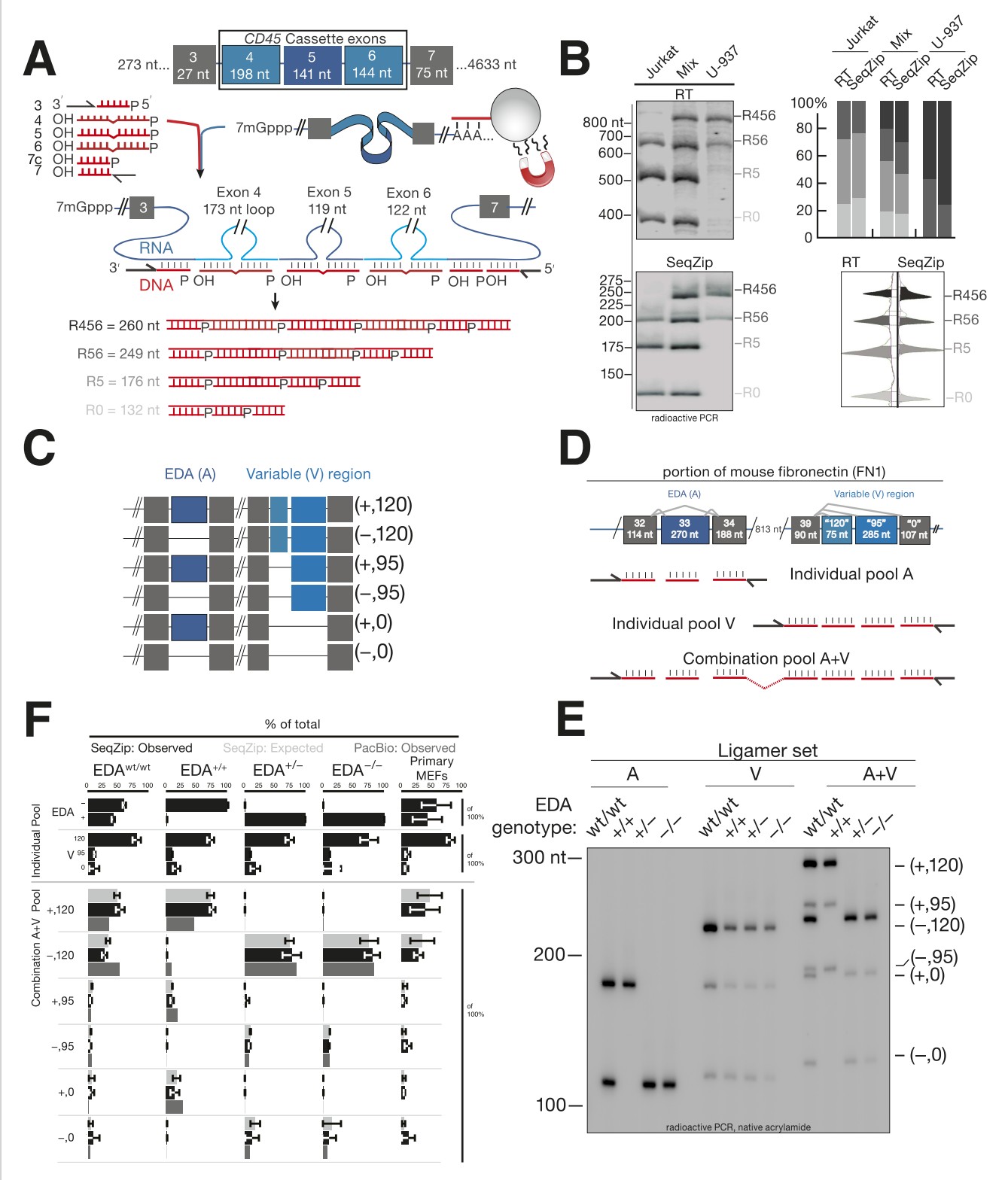

**Figure 3**. SeqZip assay to measure endogenous mRNA isoform expression. (**A**) The SeqZip strategy to detect human *CD45* mRNA isoforms. (**B**) Denaturing PAGE gels showing products of reverse transcriptase (RT) (top left) or SeqZip (bottom left) *CD45* mRNA obtained from two different human Jurkat and U-937 T-cell lines, or a 1:1 mixture of the two. Top right: quantified band intensities from gels at left. Bottom right: mirrored lane profiles from the mix lanes (RT—left; SeqZip—right). (**C**) The six possible combinations of EDA (blue; + or −) and V (light blue; 120, 95 and 0) alternative splicing within
*Figure 3. continued on next page*

*Figure 3. Continued*

mouse *Fn1* transcripts. Filled boxes depict exons, diagonal lines indicate isoform sequences not shown, and straight lines show absence of exon(s) in the final mRNA. (**D**) Detailed schematic of ligamer pools used to analyze indicated regions of *Fn1* RNA. (**E**) SeqZip ligation products from immortalized MEFs with indicated *Fn1* genotypes. Radioactive PCR separated on a native acrylamide gel. (**F**) *Fn1* isoform abundance measured by SeqZip and PacBio. Black bars indicate observed individual exon ('Individual Pool'; EDA, V) or combination frequencies ('Combination A + V pool', [EDA, V]). Shown in light gray are expected combination isoform intensities, and where available, the frequency of PacBio reads (mid-gray, lower bars).

*Figure 3D,E*), and they contained no intervening region of extensive nucleotide identity. Thus, SeqZip provided a new means to test the possibility of connectivity between *Fn1* EDA and V splicing decisions.

The effects of EDA inclusion or exclusion on V region splicing were previously tested by creating mice via homologous recombination with intronic splicing enhancers modified to favor either constitutive inclusion (+/+) or exclusion (−/−) of the EDA exon (*Chauhan et al., 2004*). That study also analyzed mice heterozygous for the modified locus (+/−) and the wild-type parental strain (wt/wt). We obtained immortalized mouse embryonic fibroblasts generated from all four mouse lines and performed SeqZip analysis (*Figure 3E,F*). Three different ligamer pools allowed us to analyze each region in isolation (individual pools A and V) or both regions together (combination pool A + V) (*Figure 3D*). EDA and V isoform ratios determined from low cycle, radioactive PCR band intensities of the A and V pool ligation products (SeqZip: Observed) were used to calculate expected EDA:V isoform abundances, assuming no interdependence between the two regions (SeqZip: Expected). We also generated cDNAs by low-cycle RT-PCR and sequenced them on a Pacific Biosciences RSII instrument (PacBio:Observed), a single molecule platform with sufficient read length to maintain connectivity between the EDA and V regions (*Sharon et al., 2013*).

In both the SeqZip and PacBio data sets, constitutive EDA inclusion or exclusion was as expected in the +/+ and −/− cells, respectively. Unexpectedly, however, we could not detect any EDA inclusion in the +/− cells despite confirming the presence of both alleles in gDNA (data not shown). Regardless, neither SeqZip nor PacBio yielded any evidence for an effect of EDA inclusion or exclusion on V region splice site choice. That is, in no case was the observed frequency of any A + V combination statistically different from the frequency expected for independent events. This was also our observation in primary mouse embryonic fibroblasts (MEFs) from wild-type mice (*Figure 3F*). Our results thus support the view that the EDA and V regions of mouse *Fn1* are spliced autonomously (*Chauhan et al., 2004*).

## SeqZip eliminates template-switching artifacts in the analysis of *Dscam1* isoforms

For the *Drosophila Dscam1* gene, alternative splicing of four blocks of mutually exclusive cassette exons (exons 4, 6, 9, and 17) can potentially produce 38,016 possible mRNA isoforms (*Figure 4A*). Previous studies suggest that all isoforms can be generated (*Neves et al., 2004*; *Zhan et al., 2004*; *Sun et al., 2013*), with all 12 exon 4 variants being stochastically incorporated in individual neurons (*Miura et al., 2013*).

Previous high-throughput methods for examining *Dscam1* exon connectivity relied on RT-PCR, a technique potentially confounded by long stretches of sequence identity in the constitutive exons separating each cluster and by sequence similarity among exon 4, 6, and 9 variants (*Figure 4B*). Long regions of sequence homology promote template switching during both RT and PCR (*Judo et al., 1998*; *Houseley and Tollervey, 2010*); this can generate novel isoforms not originally present in the biological sample. SeqZip can both dramatically reduce these regions of sequence of identity (*Figure 4B*) and introduce new exon-specific codes (*Figure 4C*). Thus, in addition to maintaining connectivity information, SeqZip both compresses sequence length and increases sequence heterogeneity, thereby greatly decreasing the potential for template switching compared to cDNAs created by standard RT or circularized cDNA approaches.

Prior to our development of SeqZip, we had attempted to use a RT-PCR-based triple-read sequencing method to determine exon connectivity between *Dscam1* alternative splicing regions 4, 6, and 9 (*Figure 4D*, *Figure 4—figure supplement 1B*, 'Materials and methods'). To measure the extent of template switching, we generated four RNA transcripts corresponding to distinct isoforms. As expected, this RT-based method detected many novel transcript isoforms containing exon

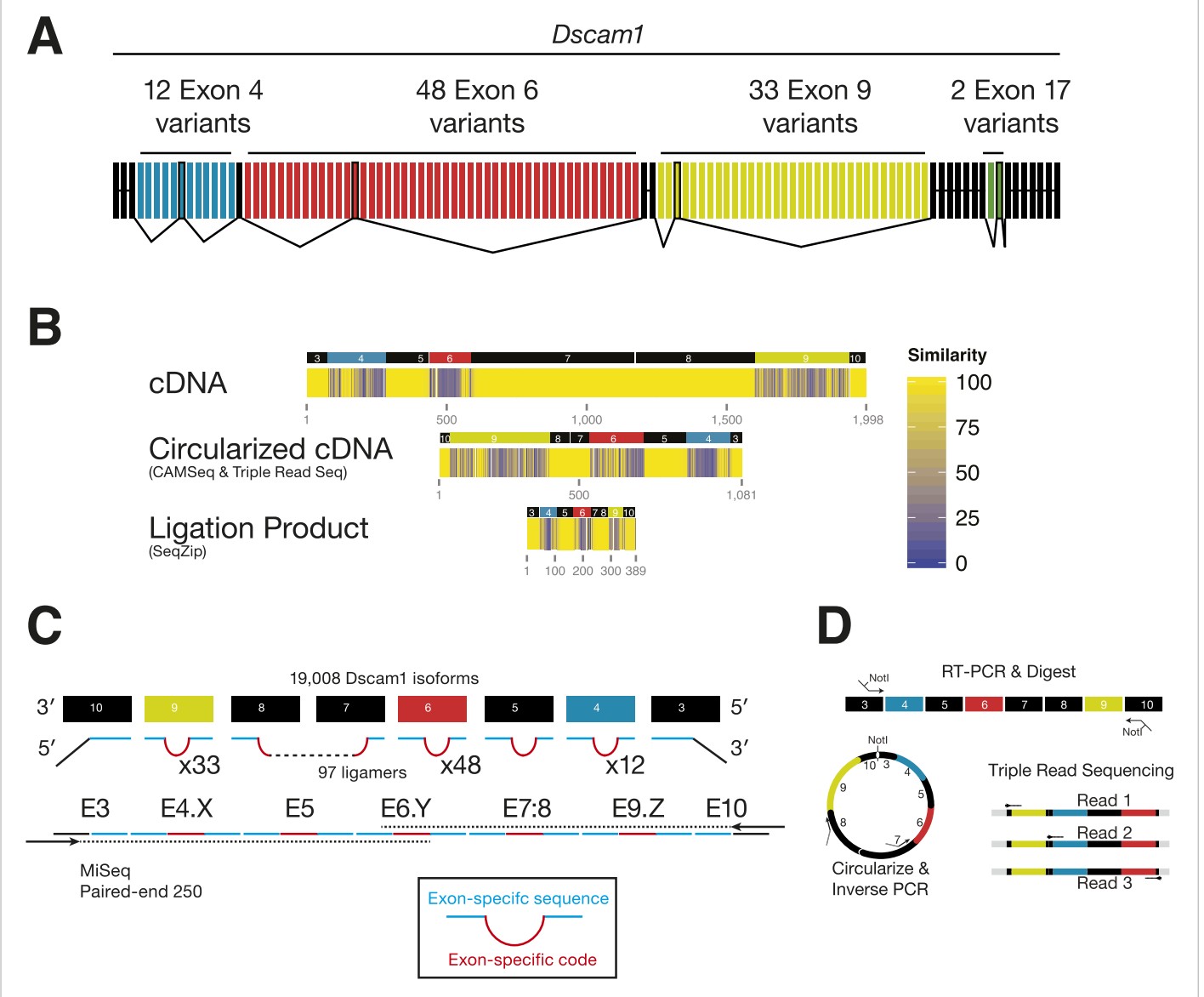

**Figure 4**. Analysis of *Dscam1* isoforms via high-throughput sequencing. (**A**) Architecture of *Dscam1*. Black: constitutively included exons; colors: variant exons. Only one cassette exon per variant region is included in the mRNA. (**B**) Sequence similarity between 1000 random isoforms of *Dscam1* in cDNA, circularized cDNA, and SeqZip ligation product form. All lengths are shown to scale. (**C**) Strategy to measure *Dscam1* isoform diversity using SeqZip on the MiSeq platform. (**D**) Strategy to measure *Dscam1* isoform diversity by triple-read sequencing on the Illumina MiSeq platform.
The following figure supplement is available for figure 4:

**Figure supplement 1**. *Dscam1* in vitro transcript measurement.

combinations not present in the four input transcripts (*Figure 4—figure supplement 1B*). These template-switched isoforms represented 34–55% of the isoforms detected, with many being significantly more abundant than one or more of the input isoforms.

A similar circularized cDNA method, CAMSeq, has also been used to assess *Dscam1* exon connectivity (*Sun et al., 2013*). In light of the high rate of template switching in our triple-read sequencing approach, we reexamined the published CAMSeq control data to assess the extent of template-switching events (*Figure 4—figure supplement 1B*). Indeed, template-switched isoforms were present in the CAMSeq data, with many template-switched isoforms being more abundant than

the low abundance input isoforms. Moreover, we detected 5386–5914 additional isoforms whose presence could not be explained by either the composition of the 8 input RNA isoforms or by template switching. Thus, while CAMSeq was a clear improvement over both previous linear RT-PCR-based approaches and our triple-read sequencing approach, template-switching artifacts remained a substantial problem.

By eliminating RT and using exon-specific barcodes to ensure unambiguous isoform assignment (*Figure 4C*), SeqZip should greatly reduce template switching. To measure this directly, we mixed together three different in vitro-transcribed *Dscam1* isoforms in the presence of total RNA from a mouse hepatoma cell line (Hepa 1–6c; *Figure 5A*). This mixture was then divided into two separate ligation reactions, each containing a complete 97 ligamer pool that differed only in the 7 nt ligamer barcode assigned to two exons in each cluster (* in *Figure 5A*). Following ligation, the differentially coded samples were mixed together, subjected to PCR, and sequenced on the MiSeq platform (on which paired-end reads can cover a total of 500 nts). Of the 50,475 reads obtained in these control reactions, none were indicative of template switching (i.e., no ligation product contained both pool 1 and pool 2 barcodes; *Figure 5B*). Moreover, when the same differential coding approach was applied to *Drosophila* S2 cell poly(A)-selected RNA, just 17 of 111,242 reads (0.015%) corresponded to template-switched isoforms (*Figure 5B*, *Figure 4—figure supplement 1B*). Thus, the SeqZip design greatly diminishes template switching.

Sequences at the ends of target exons specify where ligamers bind (*Figure 1*). The high similarity among the cassette exon sequences within each cluster raised the possibility that ligamers would bind near-cognate as well as cognate sequences. To assess the potential for mis-pairing, we calculated the free energy of hybridization (*Reuter and Mathews, 2010*) between each ligamer and all exon variants within its target cluster (*Figure 5—figure supplement 1A*). Cognate ligamer–exon pairs had predicted hybridization energies lower than $\Delta G° = -67$ kcal/mol; the closest near-cognate pair was $\geq 12$ kcal/mole higher. In the control experiments containing just three *Dscam1* isoforms, only 642 of 50,475 high-confidence alignments (1.3%) contained ligamers for exons not present in any input transcript, with the majority of these species (221/236) represented by three or fewer reads (*Figure 4—figure supplement 1B*). Nonetheless, two near-cognate hybridization products with >100 reads were detected. Although both were less abundant (2.4- and 372-fold lower) than reads corresponding to cognate targets (*Figure 5C* and *Figure 4—figure supplement 1B*), this does raise a cautionary note with regard to interpretation of extremely low abundance ligation products in experiments wherein near-cognate ligation is a possibility. On the other hand, SeqZip accurately reported input cognate isoform abundances over 3 orders of magnitude (*Figure 5C*). Thus, as with *CD45* and *Fn1* isoforms, SeqZip proved highly quantitative for assessing the majority of *Dscam1* isoforms.

## SeqZip analysis of *Dscam1* isoforms

We next used SeqZip to measure *Dscam1* isoform identity and abundance in S2 cells, as well as 4–6 hr and 14–16 hr *D. melanogaster* embryos. Ligamers targeting every exon variant in clusters 4, 6, and 9 together with ligamers for constitutive exons 3, 5, 7, 8, and 10 (97 ligamers in all) reduced the median size of mRNA sequences analyzed from 1734 nt (1722–1751 nt for exons 3–10) to 356 nt for a seven-ligamer product formed by six ligation events. This approximately fivefold length reduction allowed the products to be fully sequenced using 250 bp, paired-end reads in a single Illumina MiSeq run (*Figure 4C*). Between 449,113 and 946,110, high-confidence alignments were obtained for each sample (*Supplementary file 2*). Across all three samples, SeqZip detected 8397 of the 18,612 possible isoforms (*Figure 6A*). Individual isoform abundances were highly correlated between both technical and biological replicates ($r = 0.8–0.95$, $p < 2.2 \times 10^{-16}$, Fisher z-transformation; *Figure 6—figure supplement 1*). Of the 97 possible exons represented in our ligamer set, all were detected except exon 6.11, which is generally thought to be an unused pseudo-exon (*Neves et al., 2004*; *Zhan et al., 2004*; *Watson et al., 2005*; *Miura et al., 2013*; *Sun et al., 2013*). The absence of exon 6.11 reads from our libraries provides additional evidence for the specificity of SeqZip. Further, with two exceptions, the patterns of individual exon use in S2 cells were directly comparable between the SeqZip and CAMSeq data sets ($r = 0.87$, $p < 2.2 \times 10^{-16}$, Fisher z-transformation; *Figure 7—figure supplement 1*): exon 6.47 was well represented in the CAMSeq data but undetectable by SeqZip, and exon 9.31 was more abundantly represented in our data.

Comparison of exon usage patterns across three different biological samples revealed increasing isoform diversity with tissue complexity: S2 cells were the least diverse, 4–6 hr embryos had

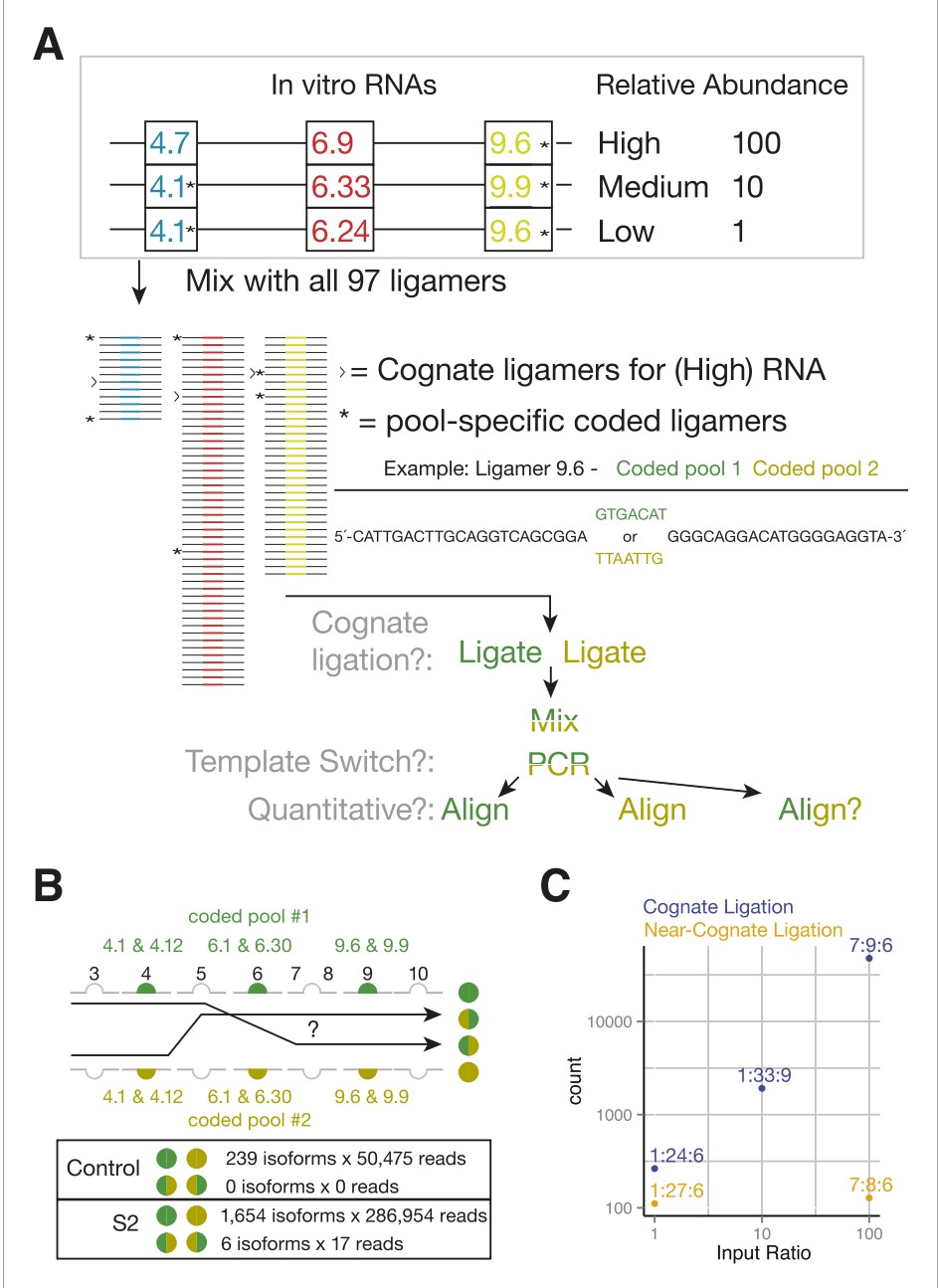

**Figure 5**. SeqZip *Dscam1* control experiments. (**A**) Three in vitro-transcribed cDNAs used as controls containing the exon variants indicated and mixed in a 100:10:1 relative ratio. Also shown are a schematic of the ligamer pool, with each ligamer targeting a different variant exon, the six ligamers (*) containing having different codes in pool 1 and pool 2, and a workflow for identifying near-cognate ligation and template-switching events. (**B**) Schematic showing how template-switched isoforms were identified as an incorrect combination of barcodes unique to the differentially coded pools shown in (**A**). Also shown are the observed numbers of un-switched and template-switched reads and isoforms for controls in (**A**) and S2 cellular RNA. (**C**) Quantification of in vitro-transcribed control cDNAs analyzed by SeqZip according to the workflow shown in (**A**).

The following figure supplement is available for figure 5:

**Figure supplement 1**. Cognate and nearest near-cognate folding energies for *Dscam1* Exon 6 variant ligamers.

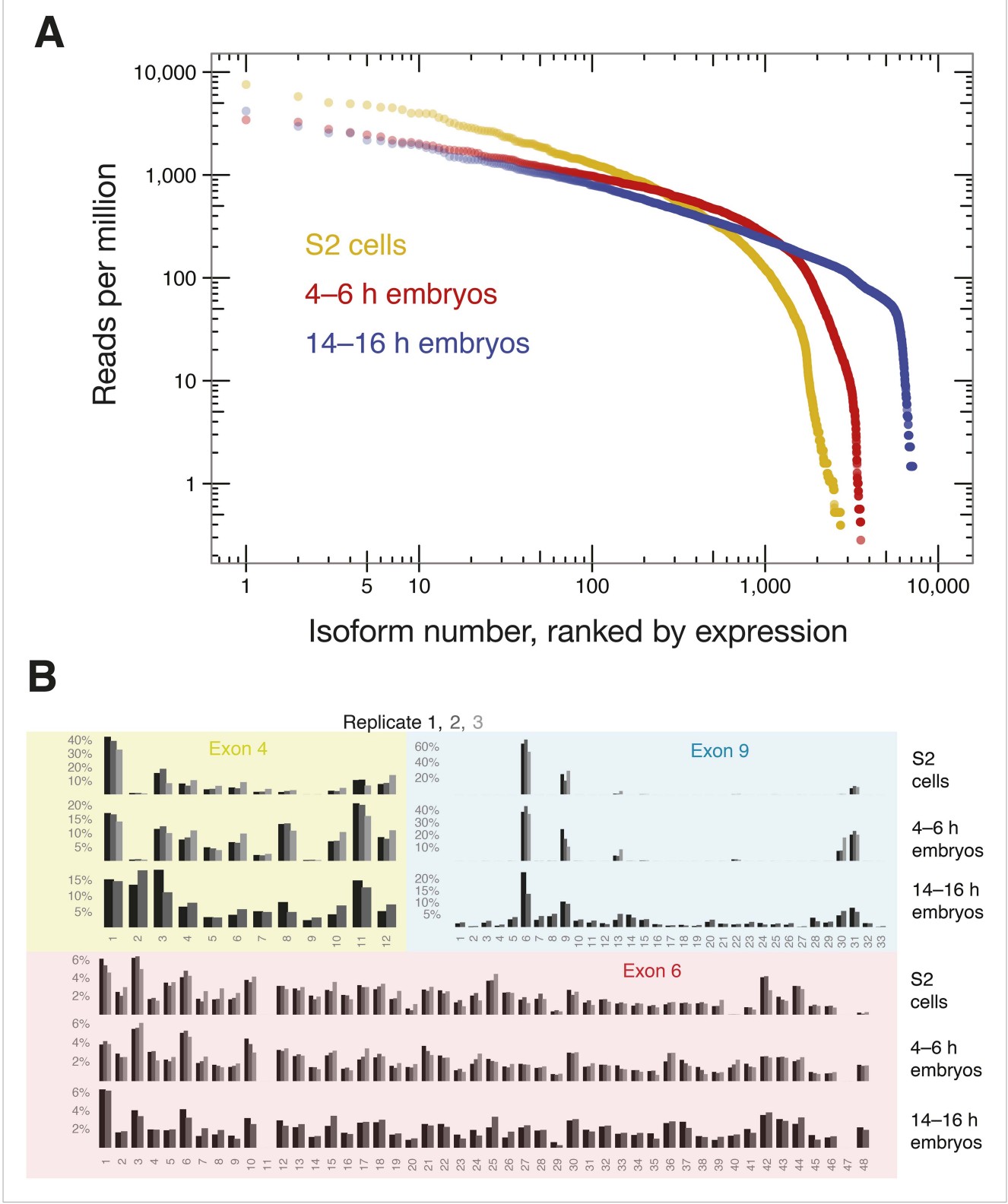

**Figure 6**. SeqZip captures diverse *Dscam1* isoform expression and exon use. (**A**) Rank-order of isoform expression by sample type (S2, 4–6 hr, 14–16 hr). (**B**) Individual exon usage per library for each replicate (differently shaded bars).

The following figure supplement is available for figure 6:

**Figure supplement 1**. Technical and biological reproducibility of SeqZip *Dscam1* isoform quantification.

intermediate isoform diversity, and 14–16 hr embryos showed the greatest isoform diversity (*Figure 6B*). As previously shown, cluster 4 and 9 exon usage patterns change during development, whereas the cluster 6 pattern remains more static (*Celotto and Graveley, 2001*; *Neves et al., 2004*; *Zhan et al., 2004*; *Miura et al., 2013*; *Sun et al., 2013*). In S2 cells, *Dscam1* mRNAs incorporate very little of exon 4 cassettes 2 and 9 and use almost exclusively exon 9 cassettes 6, 9, 13, 30, and 31. This pattern is the characteristic of hemocytes (*Watson et al., 2005*) and consistent with the macrophage-like nature of S2 cells (*Schneider, 1972*). Whereas 4–6 hr embryos are similar to S2 cells in exon clusters 4 and 9, 14–16 hr embryos show increased exon diversity, particularly in cluster 9. *Figure 7*

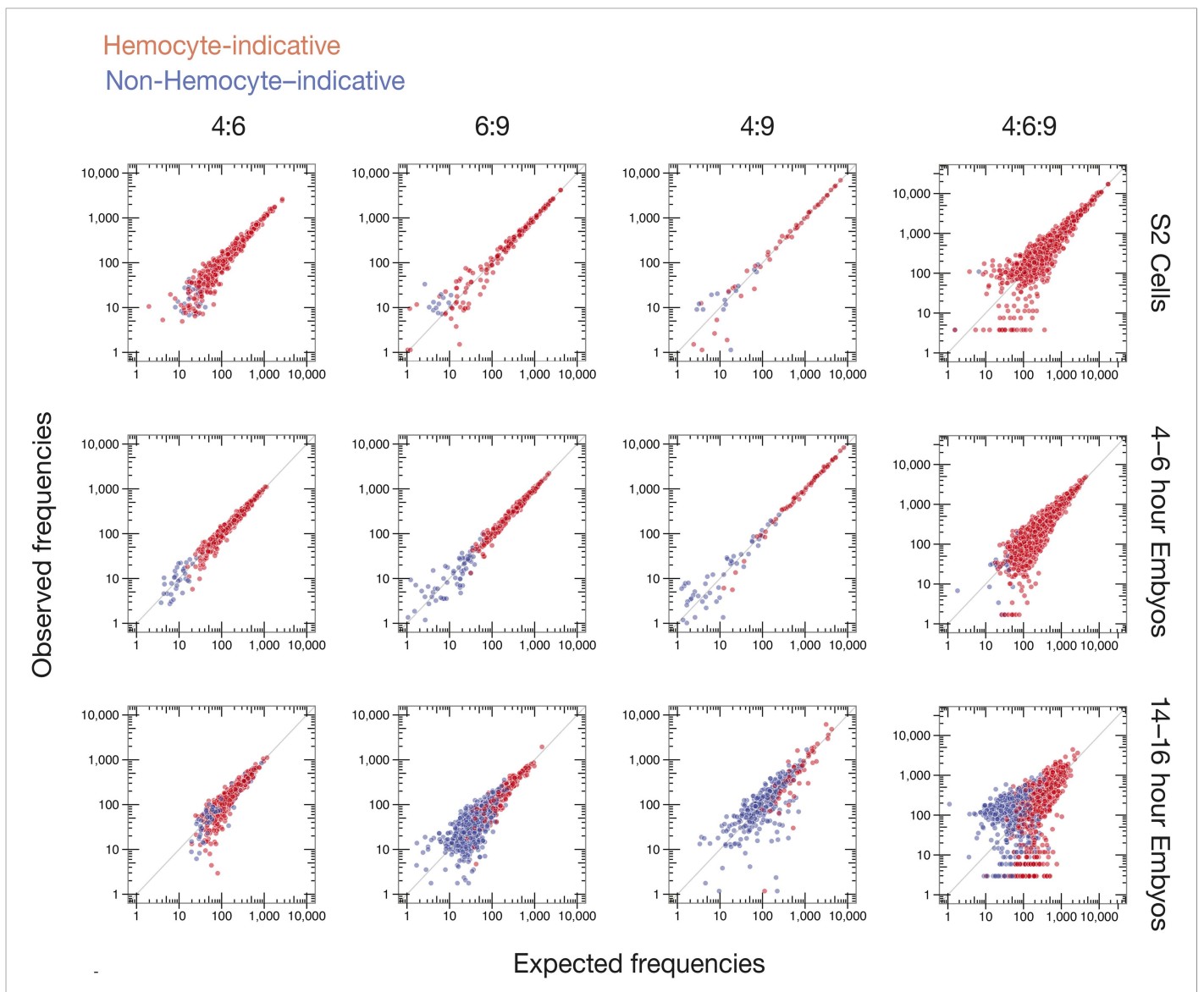

**Figure 7**. Observed vs expected *Dscam1* isoform abundance. Two-way (4:6, 6:9, and 4:9) and three-way (4:6:9) expected isoform abundances, calculated from the individual inclusion frequency for each variant exon (*Figure 6B*) in indicated sample type (S2 cells, 4–6, or 14–16 hr embryos), plotted against observed isoform abundances in that sample type. Isoforms are colored according to hemocyte-indicative (red) or non-hemocyte-indicative (blue) exon variants.

The following figure supplement is available for figure 7:

**Figure supplement 1**. Comparison of RT-PCR and ligation-based *Dscam1* isoform analysis techniques.

shows that *Dscam1* isoforms associated with hemocytes (i.e., those lacking exon 4.2 and 4.9 or containing exon 9 cassettes 6, 9, 13, 30, or 31) are the most abundant in all three samples, but other isoforms emerge as development proceeds.

To examine the possibility of coordinated splicing, we calculated expected pairwise and three-way exon combination frequencies for every transcript isoform observed in each sample (S2 cells, 4–6 hr and 14–16 hr embryos), assuming a null hypothesis of no coordination (see 'Materials and methods'). Comparison of expected and observed frequencies (*Figure 7*) revealed no statistically significant differences ($q \leq 0.05$) between expectation and observation for S2 cells and 4–6 hr embryos. For 14–16 hr embryos, however, 17 of 371 observed exon 4:9 combination frequencies, and 14 of 2004 observed 4:6:9 combination frequencies, did fall outside of the expected range ($q < 0.05$, *Supplementary file 4*). The only pattern we could deduce was that the variant 4:9 combinations were all non-hemocyte combinations (*Figure 7*). Because the majority of 4:6:9 combination frequencies (99.3%) were consistent with the null hypothesis of no coordination, our data agree with previous studies (*Neves et al., 2004*; *Sun et al., 2013*) that individual cassettes in *Dscam* clusters 4, 6, and 9 are chosen independently, with exon choice in one cluster having no detectable effect on subsequent exon choice in another cluster.

## Discussion

SeqZip provides a new strategy to retain and compress key sequence information in long RNAs while maintaining connectivity between distant regions within individual molecules. Free from the template-switching artifacts that limit RT-based methods, SeqZip can be used to quickly and efficiently examine complex alternative splicing patterns and quantify isoform expression for genes harboring multiple distal regions of alternative splicing. Here, we used SeqZip to examine the possibility of coordination between distant alternative splicing regions in mouse *Fn1* and *D. melanogaster Dscam1* mRNAs. Consistent with other studies, SeqZip revealed no evidence of such coordination.

### Coordination between alternative processing events

The idea that promoter-proximal (upstream) gene regions can affect distal (downstream) alternative splicing was first reported for the mammalian *Fn1* gene almost two decades ago (*Cramer et al., 1997*). Expressed sequence tag and oligonucleotide microarray data were subsequently interpreted to suggest coordination among different regions of alternative splicing in numerous other genes (*Fededa et al., 2005*; *Fagnani et al., 2007*). One potential explanation for this effect is promoter-dependent polymerase speed: slow RNA synthesis favors inclusion of cassette exons with weak splicing signals, while fast RNA synthesis favors their exclusion (*Fededa et al., 2005*; *Dujardin et al., 2014*). This polymerase speed effect does not necessitate any dependence of a downstream splicing decision on an upstream decision—the multiple sites of alternative splicing can be independently but similarly influenced by polymerase speed. In this case, although alternative splicing decisions in different regions within a transcript might correlate with one another, they would not depend on one another.

Neither oligonucleotide microarrays nor short high-throughput sequencing reads preserve long-range exon connectivity within individual mRNA molecules. Thus, neither approach is able to unambiguously distinguish between dependent coordination, wherein alternative processing at an upstream site causes alternative processing changes at a downstream site, and independent coordination, wherein multiple regulated exons are simply subject to similar external influences (*Calarco et al., 2007*). SeqZip, however, does preserve single molecule connectivity, so is perfectly suited to investigate coordination mechanisms. Our proof-of-principle SeqZip experiments with the mouse *Fn1* gene revealed no evidence of dependent coordination between the two regions of alternative splicing we examined (*Figure 3*). Therefore, consistent with other findings (*Chauhan et al., 2004*), we conclude that any apparent coordination between the *Fn1* EDA and V regions is due to their independent response to external influences.

### Deconvoluting *Dscam1*

With 38,016 possible isoforms, *D. melanogaster Dscam1* produces the greatest known isoform diversity of any single gene. Diverse *Dscam1* isoforms enable the developing nervous and immune systems to discriminate between heterotypic and homotypic connections (*Wojtowicz et al., 2004*; *Watson et al., 2005*; *Zipursky and Grueber, 2013*). While flies engineered to produce only 4752

unique isoforms display neurite formation functionally equivalent to wild-type controls, flies expressing just 1152 isoforms display neuronal branching defects. This supports the view that what is essential for biological function is molecular diversity not any particular sequence (*Hattori et al., 2009*). Both *D. melanogaster* and *Anopheles gambiae* (*AgDscam*) also express *Dscam* in hemocytes, where isoform diversity has been implicated in opsonizing invading pathogens (*Watson et al., 2005*; *Dong et al., 2006*).

Complete characterization of *Dscam1* isoform diversity presents an extreme technical challenge (Hattori et al., 2008). The four regions of mutually exclusive cassette exons span >4300 nt in full-length mRNAs, so maintaining connectivity among all cassettes or even just cassettes 4, 6, and 9, which span >1700 nt, is all but impossible when sequencing with current high-throughput technologies (*Black, 2000*; *LeGault and Dewey, 2013*; *Zipursky and Grueber, 2013*). Single molecule methods capable of longer reads (e.g., Pacific Biosciences) have limited read depths, making it difficult to fully analyze transcripts expressed over many orders of magnitude. Finally, many *Dscam1* exon variants arose from exon-duplication events, so their sequences are highly similar (*Lee et al., 2010*). This high-sequence similarity, combined with the long stretches of identical constitutive exons separating the distant alternative splicing regions, strongly favors template switching by RT (*Judo et al., 1998*; *Houseley and Tollervey, 2010*).

SeqZip has no RT step, it eliminates long intervening regions of common sequence, and the unique exon-specific barcodes introduced during the ligation step further discourage template switching during subsequent amplification. Using pools of 97 individual ligamers targeting every exon in clusters 4, 6, and 9, we analyzed *Dscam1* diversity in S2 cells, 4–6 hr, and 14–16 hr embryos. In all three samples, we observed individual exon use frequencies similar to those observed with CAMSeq (*Figure 7—figure supplement 1*; *Sun et al., 2013*). SeqZip and CAMSeq both detected significant exon usage changes in clusters 4 and 9 between S2 cells and embryos. Analysis of 4–6 hr and 14–16 hr embryos allowed the timing of exon 4 and 9 usage changes to be narrowed to >6 and <16 hr (*Figure 5A*), a developmental window when neurogenesis is occurring (*Goodman et al., 1993*). In S2 cells and 4–6 hr embryos, we found no evidence of inter-cluster connectivity with regard to exon choice (*Figure 7*). In 14–16 hr embryos, we found weak evidence for such connectivity (*Supplementary file 4*). Multiple cell types expressing characteristic, but different, cluster 4 and 9 exon variants, however, likely confound determination of coordination in 14–16 hr embryos. Therefore, consistent with previous reports (*Neves et al., 2004*; *Miura et al., 2013*; *Sun et al., 2013*), we conclude that individual *Dscam1* isoforms are produced via stochastic alternative splicing.

In mammalian neuronal development, cells use tandem arrays of protocadherin and neurexin genes to distinguish their own neurites from those originating from different cells. Some tandem arrays are capable of generating >1000 different spliced isoforms (*Ushkaryov et al., 1992*; *Wu and Maniatis, 1999*; *Rowen et al., 2002*). A recent analysis of mouse neurexin genes using long reads (Pacific Biosciences) of individual cDNA molecules showed that while these genes do produce many different isoforms, there is also no coordination among their alternative processing choices (*Treutlein et al., 2014*).

## SeqZip uses and limitations

A potentially routine and robust use of SeqZip is highlighted by our *Fn1* analyses, where we simultaneously measured 12 different alternative splicing isoforms and determined their relative expression by simple gel electrophoresis without sequencing (*Figure 3E*). This application is similar to the multiple-exon-skipping detection assay (MESDA) used to study *SMN1* and *SMN2* isoform expression in different Batten disease cell lines (*Singh et al., 2012*). MESDA measured the relative expression of >6 *SMN* isoforms and even identified a novel isoform, providing a useful tool for researchers working on spinal muscular atrophy. Measurement of *SMN* isoforms could also be performed using SeqZip, with several advantages over MESDA including reduction in amplicon size, lower propensity for template switching during amplification, and no RT step.

One limitation of SeqZip is the number of ligamers required to create a ligation product. To achieve necessary sequence specificity, ligamers need to be 40–60 nt. Current illumina-based sequencing platforms can read ~500 nt of contiguous sequence. Thus, ~8–12 ligamers is currently the upper limit for high-throughput sequencing analyses. Although our quantitative analysis of *CD45* showed no difference in ligation efficiency for isoforms requiring two ligations compared to those requiring five (*Figure 3B*), other transcripts could theoretically differ. Because exon-excluded isoforms

require fewer ligations, as the number of sites being examined grows, it is possible that detection of shorter (i.e., exon-excluded) transcripts will be favored. Thus, if a SeqZip experiment requires different numbers of ligation events for different RNA isoforms, it is crucial to perform the necessary controls to ensure quantitative detection of all desired isoforms. In our experiments, we were able to demonstrate accurate isoform abundance reporting over >4 orders of magnitude for *Dscam* (*Figure 5B*). At the low end of this range, however, near-cognate ligation events began to be problematic. As for sensitivity, we have been able to obtain detectable ligation products from as few as ∼900 ($5 \times 10^{-17}$ M; 0.05 fM) target RNA molecules (data not shown). Because the limit of detection for SeqZip is likely more than a single molecule, lack of detection of a particular isoform should only be interpreted as that isoform being below the SeqZip detection limit. Nonetheless, when properly controlled, SeqZip is a sensitive quantitative method for assessing complex isoform abundance patterns over a wide dynamic range.

The easiest 'complex' form of alternative splicing for SeqZip analysis was *Dscam1*, where alternative processing is limited to mutually exclusive exons (i.e., all spliced isoforms contain the same number of exons, and therefore, ligamer ligation events). However, many mammalian transcripts have more varied alternative splicing, including alternative 5′ and 3′ splice site usage and intron retention. Ligamer design against these types of alternative splicing quickly becomes unwieldy. For example, characterization of alternative transcriptional start and polyadenylation sites requires different terminal ligamers for each different start or polyadenylation site. Thus, while we were able to simultaneously assess two different types of alternative splicing in *Fn1* (exon inclusion/exclusion and alternative 3′ splice sites), other mammalian genes displaying even more numerous forms of alternative processing (e.g., *Kcnma1*) would require significantly more complicated ligamer pools. Indeed, there may be genes with splicing patterns that cannot be readily interrogated with a single ligamer pool capable of generating a unique ligation product for every possible isoform. In such cases, analysis using multiple ligamer pools each targeting a select region may still be advantageous for estimating splicing frequencies compared to more traditional methods like quantitative RT-PCR. To assist readers in designing their own ligamer pools, we have included a schematic of our ligamer design process for mouse *Fn1* (*Figure 1—figure supplement 1* and *Supplementary file 3*). For highly complex pools, this process can be automated by writing a simple Python, Perl, or R script specific to the problem being addressed.

One potential future application of SeqZip is the detection of multiple single-nucleotide polymorphisms (SNPs) on a single molecule of a long RNA. By placing the ligation sites over each SNP, one could take advantage of the requirement by Rnl2 for complete complementarity at a ligation junction; mismatches would inhibit efficient ligamer joining (*Landegren et al., 1988*; *Chauleau and Shuman, 2013*). Further, any sequence can be placed in between the two regions of target complementarity within each ligamer. Therefore, sequences for custom priming, restriction digestion, recombination, etc, can be introduced, allowing for quantification or subsequent manipulation of ligation products. Analysis of ligation products can even be multiplexed, allowing for simultaneous generation and analysis using internal controls. These applications and others are shown in *Figure 1—figure supplement 2*.

As demonstrated by our investigation of *Dscam1*, SeqZip ligation products can be analyzed by high-throughput sequencing via incorporation of platform-appropriate priming sequences in the terminal ligamers or PCR primers or in the spacer sequences of internal ligamers. SeqZip could also be used to assess the integrity of very long RNAs, such as piRNA-precursor transcripts (*Li et al., 2013*) or mRNAs with extended 3′ UTRs (*Wang and Yi, 2013*). Thus, SeqZip, which retains sequence connectivity and overcomes template-switching artifacts of RT-based methods, represents a useful and adaptable new tool for detecting and quantifying numerous features of individual molecules of long RNA.

## Materials and methods

All oligo and ligamer sequences are provided in *Supplementary file 1*. U-937 (CRL-1593.2), Jurkat (TIB-152), and S2 (CRL-1963) cell lines were from ATCC. Primary C57BL/6J MEF cells were from Jackson Labs. MEF lines were immortalized using SV40 retroviral infection. *D. melanogaster* embryos were reared at 25°C.

### Proof-of-concept experiments

The template sequence for the initial ligase screen (*Figure 2A*) was a 307 nt section of mouse DDX1 mRNA (NM_134040.1; see *Supplementary file 1*); ssDNA and RNA templates were a synthetic oligonucleotide and in vitro transcript, respectively. Enzymes tested were *Tth* DNA ligase (AB-0325;

Thermo, Waltham, MA), *Tsc* DNA ligase (Dlig 119; Prokaria, Reykjavik, Iceland), thermostable DNA ligase (BIO-27045; Bioline, Taunton, MA), T4 DNA ligase (M0202S; NEB, Ipswich, MA), *Escherichia coli* DNA ligase (M0205S; NEB), Rnl2 (M0239; NEB), SplintR ligase (M0375; NEB). Templates and oligos were hybridized by heating to 65°C for 1 min, followed by slow cooling to room temperature. After buffer and enzyme addition, reactions were incubated at the manufacturer-specified optimal ligation temperature (16–65°C depending on enzyme) for time indicated; denaturing polyacrylamide gels were quantified by phosphorimaging. Final ligation conditions in *Figure 2A* were (left panel) 1.5 µM ssDNA or RNA template, 5′-$^{32}$P-labeled oligos (10 µM each), and 1 µl of neat indicated enzyme (specific units varied according to the manufacturer and enzyme) in manufacturer's recommended buffer; (right panel) 250 nM RNA template, 5′-$^{32}$P-labeled oligos (500 nM each), and 10 U/µl Rnl2 or 20 U/µl T4 DNA ligase. Reactions in *Figure 2B* contained 1.25 µM *DDX1* RNA template, 5 µM each 5′-$^{32}$P-labeled oligo, and 10 U/µl Rnl2 were incubated for 4 hr at 37°C and separated on a 11.25% denaturing polyacrylamide gel. Reaction conditions in *Figure 2C,D* were as described in the SeqZip section (see below), using indicated RNA templates in a background of 10 ng/µl total mouse embryo fibroblast (MEF) RNA. RNA templates in *Figure 2C,D* were runoff transcripts from PCR products generated with different oligo combinations (*Supplementary file 1*) having partial complementarity to human eIF4A3 cDNA (RefSeq: NM_014740).

## Radioactive and end point PCR

For radioactive PCR using Taq Polymerase (PN-M712; Promega, GoTaq Green Master Mix, Madison, WI), one PCR oligo was $^{32}$P-5′-end-labeled, and cycle numbers were confined to a range predetermined to yield a per cycle log2 linear increase in signal intensity (typically 15–23 cycles). After resolution on a denaturing polyacrylamide gel, bands were quantified using a Typhoon imager (GE Healthcare, Chicago, IL) and the ImageQuant software package (GE Healthcare). End point PCR was typically 35 cycles at a hybridization temperature 5°C below the lowest primer $T_M$. End point PCR products were resolved on native 29:1 (acrylamide: bis-acrylamide) polyacrylamide gels, visualized by staining with SybrGold (Invitrogen, Grand Island, NY), and also imaged/quantified as above.

## Ligamer design

For SeqZip of endogenous *CD45*, *Fn1*, and *Dscam1* mRNAs, individual ligamers were designed as follows (*Figure 1—figure supplement 1*). The 5′- and 3′-termini of each target sequence (e.g., one or multiple exons) were obtained from online databases (Ensembl and UCSD). For terminal ligamers, the length of complementarity necessary to obtain a predicted hybridization Tm nearest but not exceeding 65°C was calculated using the BioPerl Bio::SeqFeature::Primer Tm module with default [Na+] and [oligo] settings (*Allawi and Santalucia, 1997*). This complementary sequence was then appended to the desired PCR-primer hybridization sequence. For internal ligamers, the length of complementarity at each end (generally 12–25 nt) was adjusted to achieve a Tm of 60 ± 5°C for each end separately in order to maintain an overall length of ≤60 nt. End sequences were joined via a short spacer that could include a barcode. Ligamers were ordered from Integrated DNA Technologies, with or without a 5′ phosphate as required, and used directly in SeqZip reactions.

## SeqZip

Total RNA (200–800 ng per ligation reaction) isolated from cells using Tri Reagent (MRC) was bound to Poly(A)Purist MAG magnetic beads (Ambion AM1922; 2.25 µl slurry per ligation reaction) according to the manufacturer's instructions. After removal of unbound RNA, ligamers (10 nM (f.c.)) were hybridized to bead-bound poly(A) RNA in hybridization buffer (60 mM Tris-HCl, pH 7.5 at 25°C, 1.2 mM DTT, 2.4 mM MgCl$_2$, 480 µM ATP) by heating samples to 62°C for 5 min in a thermocycler and then slow cooling to 45°C via a 3°C drop every 10 min. After 1 hr at 45°C, the temperature was again decreased 3°C every 10 min to 37°C, where samples were held until T4 RNA ligase 2 (2 U/µl (f.c.), NEB, M0239) addition. At this point, the samples were in 1× ligation buffer (51 mM Tris-HCl, pH 7.5 at 25°C, 2 mM DTT, 5 mM KCl, 2 mM MgCl$_2$, 400 µM ATP, 3.5 mM (NH$_4$)$_2$SO$_4$, 5% (vol/vol) glycerol). After 8–16 hr at 37°C, beads were used directly for PCR using a polymerase appropriate for the downstream application (e.g., Taq for gel analysis, Herculase for sequencing).

## Reverse transcription and PacBio FN1 analysis

RT reactions in *Figure 3* used SuperScript III (10 U/µl, Invitrogen) at 55°C, 200 ng poly(A) selected RNA, and either anchored oligo(dT) or a gene-specific antisense primer. *Fn1* amplicons were

prepared using 12 cycles of Herculase II Fusion DNA Polymerase and primers targeting the sequence between the EDB and V regions (*Supplementary file 1*). Amplicons were submitted for library construction using The DNA Template Prep Kit 2.0 (Pacific Biosciences) and sequenced on a PacBio RS II. Circular consensus reads were aligned to an index of FN1 isoforms using BLAT.

## Triple-read sequencing

RT was performed using 5 µg total RNA, Superscript II (Invitrogen), and random hexamers at 42°C for 1 hr. Strand-switching control experiments were performed by mixing plasmids encoding *Dscam* isoforms 1.33.9, 12.32.9, 1.24.6, and 7.9.6 at 3:3:1:1, 1:1:1:5, and 1:1:1:1. PCR with Phusion polymerase (NEB) (annealing temperature, 55°C; 1 min extension) was used to amplify cDNA or plasmids containing the region encompassing exons 4, 6, and 9 with exon 3 (Not1Ex3For: TAT CGG CGG CCG CGG ACG TCC ATG TGC GAG CCG) and exon 10 (Ex10RevNot1: ATA TCG CGG CCG CGA GGA TCC ATC TGG GAG GTA) primers. Both primers contained a 5′ end *Not*I restriction site. PCR products were gel purified and digested with *Not*I for 2 hr at 37°C, followed by a heat inactivation at 65°C for 20 min. The digested PCR products (0.5 µg) were circularized in 500 µl 1× T4 DNA ligase buffer (NEB) with 1 µl T4 DNA ligase (0.8 U/µl, (f.c.), NEB, M0202) at 18°C overnight. Inverse PCR was then performed using Phusion polymerase (annealing temperature, 55°C; 30 s extension) with primers specific to exons 7 (PEex7Rev: CAA GCA GAA GAC GGC ATA CGA GAT CGG TCT CGG CAT CCC TGC TGA ACC GCT CTT CCG ATC TAT GAA CTT GTA CCA T) and 8 (PEex8For: AAT GAT ACG GCG ACC ACC GAG ATC TAC ACT GTT CCC TAC ACG ACG CTC TTC CGA TCT AAG TGC AAG TCA TGG) that contained Illumina paired-end clustering sequences. Libraries were gel purified, quantified via Nanodrop (Thermo), and clustered on a Genome Analyzer IIx (GAIIx) paired-end flow cell on an Illumina cluster station using the standard clustering protocol.

Sequencing was performed on an Illumina GAIIx by modifying the protocol for paired-end sequencing with an index read. Briefly, read 1 was performed for 24 cycles with a primer complementary to the 5′ end of exon 8 (Ex8For: ACG ACG CTC TTC CGA TCT AAG TGC AAG TCA TGG). The flow cell was denatured to remove the exon 9 sequencing products, a primer complimentary to exon 3 (Ex3For: CCC GGG ACG TCC ATG TGC GAG CCG) was annealed, and read 2 sequenced for 12 cycles. Next, the flow cell was re-clustered using the paired-end protocol, and read 3 performed for 20 cycles using a primer complementary to exon 7 (Ex7Rev: GAA CCG CTC TTC CGA TCT ATG AAC TTG TAC CAT).

Base calling was performed from the raw images using the Firecrest, Bustard, and Gerald software modules of GAPipeline-1.4.0 and a matrix.txt file for a PhiX lane from a previous flow cell for calibration. This generated a single FastQ file per lane containing the three catenated reads from each cluster. The reads within the FastQ files were parsed to separate the three reads, the identity of each exon determined, and then the full isoform determined by matching to a database of known exon sequences.

## MiSeq library preparation

SeqZip ligation reactions were amplified via PCR (Agilent, Herculase II Fusion DNA Polymerase, Catalog Number—600675) for 12 cycles using common primers. Reactions were resolved on a 5% polyacrylamide native gel, and DNA in the size range appropriate for full-length ligation products quantified by fluorescence imaging, cut and eluted from the gel, and precipitated. Equal DNA quantities based on the gel imaging were amplified for another 22 cycles using primers containing Illumina priming sequences with integrated barcodes. PCR products were purified (28104; Qiagen, QIAquick PCR Purification Kit) and quantified using a Bioanalyzer 2100 (Agilent) and High-Sensitivity DNA chip. Samples were mixed and submitted for sequencing on the MiSeq instrument using the paired-end 250 nt read option. Sequencing data are available at Short Read Archive accession SRP043516.

## MiSeq read analysis

All Dscam1 SeqZip products were shorter than 400 nt; therefore, paired-end MiSeq 250 nt reads contained overlapping 3′ sequences. Using these overlapping sequences, paired reads were combined into one sequence using the Paired-End Assembler (pear, v. 0.8.1) and default options (*Zhang et al., 2014*). An index of all possible Dscam1 ligamer combinations was created using a single PERL script that permuted all possible ligamer combinations with correct 5′ to 3′ exons 4, 6, and 9

ligamer arrangements. Paired MiSeq reads were aligned against this index using Bowtie2 v. 2.1.0 (*Langmead and Salzberg, 2012*) in the very-sensitive-local mode and constrained using no-discordant to only look for reads where both pairs aligned to the same isoform. Using the SAMtools (v. 0.1.19) software package (*Li et al., 2009*), alignments were further filtered for alignments containing quality 31 and above (-q 31) and read counts per isoform extracted. Count analysis was performed and graphs generated using R (*R Development Core Team, 2008*).

## Differential expression of *Dscam1* isoforms

For differential expression analysis, we treated each *Dscam1* isoform as though it was its own gene. The percent use of individual exons for each cluster (4, 6, and 9) was determined. Expected use of all possible combinations of 4:6, 4:9, 6:9, and 4:6:9 was calculated by multiplying the percentages of individual exon use. Expected use was compared to observed use of the equivalent combination. The DESeq differential gene expression R package (*Anders and Huber, 2010*) was used to identify isoforms whose observed and expected abundances were 'differential'.

## Determining sequencing similarity of *Dscam1* sequences

Endogenous *Dscam1* sequences were obtained from genomic build DM3 using BEDTools (*Quinlan and Hall, 2010*). All possible *Dscam1* isoform sequences between exons 4 and 10 were assembled using a PERL script. Five hundred random isoforms were obtained and aligned using TCOFFEE (*Di Tommaso et al., 2011*) in the Jalview package (*Waterhouse et al., 2009*). Consensus scores of alignments were exported and graphed in R. The same analysis was performed on *Dscam1* ligation products, except ligamer sequences were used in place of endogenous exonic sequences.

## Statistical analysis

Error bars represent the standard error of the mean of experimental replicates. Errors were propagated from individual standard deviations according to standard methods (*Goodman, 1960*; *Natrella, 2012*).

## Acknowledgements

This work was supported in part by National Institutes of Health grants R01 GM067842 to BRG and GM62862 to PDZ. We thank Alper Kucukural for bioinformatics support, Kristen Lynch for plasmids, Andrés Muro for *Fn1* cell lines, and members of the Moore and Zamore labs for helpful comments on the manuscript.

## Additional information

### Competing interests

PDZ: Reviewing Editor, *eLife*. The other authors declare that no competing interests exist.

### Funding

| Funder | Grant reference | Author |
|---|---|---|
| National Institutes of Health (NIH) | GM62862 | Phillip D Zamore |
| National Institutes of Health (NIH) | GM067842 | Sara Olson, Brenton R Graveley |

The funder had no role in study design, data collection and interpretation, or the decision to submit the work for publication.

### Author contributions

CKR, Conception and design, Acquisition of data, Analysis and interpretation of data, Drafting or revising the article, Contributed unpublished essential data or reagents; SO, Acquisition of data, Contributed unpublished essential data or reagents; BRG, Acquisition of data, Analysis and interpretation of data, Drafting or revising the article, Contributed unpublished essential data or reagents; PDZ, Conception and design, Analysis and interpretation of data, Drafting or revising the article; MJM, Conception and design, Analysis and interpretation of data

## Additional files

### Supplementary files

- Supplementary file 1. Ligamer sequences and other information.

- Supplementary file 2. Sequencing statistics.

- Supplementary file 3. Ligamer folding energies.

- Supplementary file 4. *Dscam1* isoforms with observed expression significantly different from expected.

### Major dataset

The following dataset was generated:

| Author (s) | Year | Dataset title | Dataset ID and/or URL | Database, license, and accessibility information |
|---|---|---|---|---|
| Roy C | 2014 | Assessing long-distance RNA sequence connectivity via RNA-templated DNA-DNA ligation | http://www.ncbi.nlm.nih.gov/sra/?term=SRP043516 | Publicly available at NCBI Short Read Archive (SRP043516). |

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
