## [Decision Letter]

Thank you for sending your work entitled “Assessing long-distance RNA sequence connectivity via RNA-templated DNA-DNA ligation” for consideration at *eLife*. Your article has been favorably evaluated by Aviv Regev (Senior editor), a Reviewing editor, and three reviewers.

Roy and colleagues present a new method for the assessment of complex alternative splicing systems. Most current methods for profiling the expressed alternatively spliced isoforms of a gene use short read RNAseq to assess the linkage of individual exons and thus require different positions of alternative splicing within a transcript to be examined independently. This is a problem for assembling the correct set of complete mRNAs from a gene as information on the combinatorial relationships and possible coregulation of distant exons is lost, although these relationships can frequently be deduced by statistical inference. The new method, “SeqZip”, uses oligo ligation directed by the mRNA template to report on exon-exon joints present in the mRNA and to condense distant regions of splice variation into a much shorter sequences assayable by sequencing or simply by size. The method is clever and the biochemistry is very well done, but no new biological insights are presented. The interest here is in the technology and the widespread need for such an assay. However, as a methods paper the study has several failings that need to be addressed.

Major points:

1) From the data presented, it is not clear how well a ligation event for an oligo pair will linearly report on a particular exon-exon joint when the pair is part of a complex set of ligamers. This has several complications.

A) From Figure 2, it appears that increasing the total number of ligation events decreases the efficiency of product generation. This will lead to a favoring of exon skipping events in the assay. In Figure 3, the authors compare the exon inclusion ratios of *CD45* measured by RT/PCR and SeqZip and find that they are similar. However, RT/PCR is known to be biased towards exon-skipped products, especially when comparing size differences as large as those here. Do the similar results from the two assays indicate a similar exon-skipping bias from SeqZip?

B) Another source of error in the assay will likely be differences in annealing and ligation efficiency between different ligamer pairs. One oligo can have quite different ligation activities when joined to different partners. Different oligos have such drastically different annealing and ligation efficiency especially since they can also form various inter- and intra-molecular interactions when the complexity is increased. As a result, the poorest one will be the rate-limiting factor for the whole isoform presentation. For individual isoform pairs under different conditions, we use ratio/ratio to infer changes, but in the current study, this is not feasible. They may argue that a poor oligo will affect all isoforms equally, and thus, it is not a problem. However, other isoforms that lack the targeting exon for such poor oligo will be over represented. This problem may not surface with low complexity system. But, it is likely that it will soon be encountered by others, if the system is adopted, and if not properly addressed, the method will have little impact. This is a critical point for the authors to assess.

C) Related to B, they make passing reference to matching the free energy of binding for each hybridizing segment, but it is not clear how well this works or whether it makes all oligo pairs join equally well. Similarly, the authors discuss the question of distinguishing highly similar exon sequences in DSCAM, but it is not clear if their approach is a general solution. Through assays of constructed pools they show they can distinguish exons 4.1 and 4.7, exons 6.9, 6.33, and 6.24, and exons 9.6 and 9.9, using their complex set of ligamers. However, it is not clear if these are the most similar of all DSCAM exon pairs or that others of the many possibilities might be more difficult to distinguish.

2) Although the authors present a complex oligo set for DSCAM whose ligated product can be assayed in a length accessible by MiSeq, it is not clear how general these conditions are for other complex systems, where there are more sites of variation and more types of splicing change. It seems likely that some regions will still need to be assayed separately and combinations predicted statistically. What are the limitations here? How complex a set of ligamers is feasible? Is there an upper limit on length that can be sequenced as a single fragment? Given such a length limit, what determines the number and spacing of alternative events that can be assayed by a ligamer set targeting all the possible products? Exactly how did the DSCAM mapping and variant quantification work? It appears that DSCAM isoforms were identified and counted by aligning to a library of isoform sequences and somehow scoring for sequence match. This needs to be described in much more detail. For this to be a useful method, it will need to be applicable to other complex transcript sets. It is not clear how feasible this is. Without understanding the mapping strategy, it is not clear how one might develop this for another gene.

Minor points:

1) The paper is very haphazardly written and it is often difficult to follow exactly what is being done. At the same time, the authors often overstate the novelty and relative performance of the assay. Oligo Ligation as an assay for specific sequences in a complex mixture goes back at least to Landegren and Hood. The method here is basically an extension of RASL-seq. It doesn't take away from the advance to reference these precedents. The authors comment on the better performance on SeqZip over CamSeq and PacBio long read sequencing. However, examining Figure 2 and Supplement 6A, they appear comparable. When discussing CamSeq, in the section “SeqZip eliminates *Dscam1* isoform template switching analysis artifacts”, it is not clear what the authors mean by the sentence: “Although template-switched and unexplained isoforms together represented just 0.6-0.94% of all reads, they account for 99.9% of multi-exon isoforms sequenced…” The authors state that detection of template switched isoforms is much lower for SeqZip, but later report that 1.3% of the alignments in the sequencing of the DSCAM ligation products contained exons not present in the input. From this it would seem that the error rate for SeqZip is potentially higher than CamSeq.

2) It is often unclear what the figures are presenting. More labels and lane numbers are needed. In Figure 2, it was not clear that each bar represented a different variant for each region and should thus sum to 100%. In this Figure, the EDA bars for the EDA^+/-^ cells show only one isoform (they are not labeled as to which is which). Shouldn't both isoforms be present in these cells? The point of Figure 4 is also not clear.

3) As mentioned above, how were the isoform abundances calculated? This should be described in detail with a figure showing the conversion from read counts to isoforms. Also needed are the data tables and sequencing statistics for the MiSeq runs. Exactly how were these sequences mapped to transcripts to identify the exon combinations? The authors report the use of DESeq to determine differential expression of DSCAM isoforms. They should be aware that this is considered to be a bad idea generally and specifically that SeqZip libraries likely violate several crucial assumptions made by DESeq. This should be clarified in detail.

4) The authors stress that PCR amplification must be minimized, but report 12 cycles of PCR after ligation, followed by 22 cycles to add the Illumina primers. How were these numbers arrived at? Was it the same for the biological samples and the artificial pools? How will this be affected by transcript abundance for targets that are less abundant than DSCAM? The DSCAM splice variants are all the same length, but this is not usually the case for systems of complex splicing. How will this PCR step affect the quantification when some variants are much shorter than others?

[Editors' note: further revisions were requested prior to acceptance, as described below.]

Thank you for resubmitting your work entitled “Assessing long-distance RNA sequence connectivity via RNA-templated DNA-DNA ligation” for further consideration at *eLife*. Your revised article has been favorably evaluated by Aviv Regev (Senior editor), a member of the Board of Reviewing Editors, and the original reviewers. The manuscript has been improved, but all the reviewers feel that there are some comments that would require a minor revision. Once these are introduced to the description of the results, methods and/or discussion (no further experiments or analyses are requested) we will be able to accept the paper. We emphasize that we do not anticipate re-sending the manuscript to the reviewers.

Specifically:

1) The authors should add a discussion of areas for future analytics to better improve the reliability and quantifiability of the inferred isoforms. These should accompany the currently well-described experimental limitations of SeqZip. As Reviewer 1 notes in further detail (and agreed by all reviewers): (1) there may be genes and isoforms that cannot readily be assayed with ligamer pools that uniquely quantify combinations of splicing events, but proper inference would still be able to help estimate splicing frequency; (2) guidelines for a method/algorithm to design a valid ligamer pool for a given gene are important to highlight in terms of what is already available and what is a future analytic direction.

2) Several critical details are missing or quite unclear, especially in the Methods sections, which were missing before or even removed in this version, per the items detailed in Reviewer 2's review. In particular, we believe that the information on number of PCR cycles is *absolutely critical* to reinstate to the Methods section.

3) There are some lingering concerns with the extent to which the method is (1) quantitative and (2) has false negatives (the two are inter-related). Again, we would like to see the discussion section clarify to the reader, as much as possible, what they can and cannot expect from the method. This is important for its long-term impact in the field.

*Reviewer 1*:

The authors have clarified a number of concerns I had with this revised version of the manuscript. The authors' section on the limitations of SeqZip clearly specifies most of the experimental limitations that remain. I would have liked to see a more thorough discussion of the analytic limitations or areas for improvement in the Discussion. For example:

1) As the authors note, there may be genes and isoforms that cannot readily be assayed with ligamer pools that uniquely quantify combinations of splicing events. However, SeqZip could still provide major advantages over conventional RNA-Seq for these genes, if it were coupled with statistical inference to estimate the frequency of splicing event combinations. The requirement that reads uniquely quantify isoforms is helpful but probably unnecessary for recovering accurate values. Using DESeq will only work well with ligamer pools that uniquely specify splicing combinations. For more complex pools, the authors’ current pipeline is likely to return unreliable numbers.

2) The authors do not describe an algorithm that generates a valid ligamer pool for a given gene. How are users of SeqZip to know that the pool they design meets the requirements of the assay and the analysis pipeline? What if they make a mistake, and leave a particular combination of splicing outcomes “uncovered” by the ligamer pool? A rigorous design tool for making ligamer pools would ensure that the current analysis pipeline returns valid quantification. Alternatively, the analysis pipeline itself could be improved (as noted in #1) to support more relaxed ligamer designs.

Both of these issues seem like good potential future directions. I would just suggest that the authors note them as limitations or opportunities in the Discussion.

*Reviewer 2*:

While the authors addressed most of the concerns raised earlier, this “final” version still has various problems:

1) The optimal PCR cycle number. The authors argued that under their conditions, the PCR cycle up to 22 did not cause any problem, yet still recommended to keep the cycle minimal in the previous version. Now, the authors removed such recommendation because they no longer think it is a critical issue? They did not even specify in their method section how many PCR cycles used. This information is needed for others to use the method.

2) Template switching was not a major problem with the amount of starting materials the authors used. However, most experiments use total RNA from a large number of cells. The authors need to provide instruction in their Methods section for the optimal range of RNA concentrations to be used in their assays.

3) I am still confused by the phrase in the section “SeqZip eliminates template-switching artifacts in the analysis of *Dscam1* isoforms”: “Although the percentage of total reads aligning to template-switched and unexplained isoforms was low (0.6-0.94%), these few reads support the majority of *Dscam1* isoforms sequenced (99.9%)”. Did the authors mean the majority of artifacts sequenced? If this is what they meant, SeqZip detected 1.3% artifacts, which is higher than CAMSeq.

4) In the Discussion, the authors cited “slow RNA synthesis favors inclusion of cassette exons with weak splicing signals, while fast RNA synthesis favors exclusion (Kornblihtt et al. 2013)”. They need to cite a recent paper in Mol Cell from the lab that has corrected this simple version of the model.

5) The authors added additional references in the text, but many did not show up in the reference list. Seem to need to update their EndNote. They should cite the Hood Science paper, which is the first to describe the ligation approach.

*Reviewer 3*:

The SeqZip method presented by the authors does address an important issue that is of general interest. The concept for the method has been around previously. Thus, I consider it essential that a high-profile publication reporting this method contains extensive validation of the approach.

The authors have made an attempt to address the issues raised in the review. However, the major issue remains: under which circumstances can the SeqZip method be considered as quantitative? To ensure the quantification aspect of SeqZip, several conditions have to be examined:

i) the specificity of the ligamers;

ii) the equal hybridization and ligation efficiencies according to ligamer sequences;

iii) the equal ligation efficiencies across various numbers of ligation events.

These key parameters differ for each target and rigorous control criteria are essential to set up suitable conditions for each target. Without prior validation of these criteria, SeqZip cannot be considered as quantifiable. The only conclusion that could thus be addressed is that combination of alternative events is present within one transcript if there is a signal (but absence of signal doesn't necessary mean that the isoform doesn't exist).

The authors have selected different targets to validate different aspects of the method (*CD45*, *FN1* and *Dscam1*) but their major figure describing SeqZip method (Figure 2) highlights the strong bias to underrepresent mRNA isoforms that require multiple ligation events.

The Figure 9 increases the concern that also ligamers can be non-specific. Nevertheless, the authors strongly argue against relying on the CAMseq experiments, which at the same time they consider as not reliable. Again, a more rigorous, systematic inclusion of controls is needed. For example, the authors should have selected control constructs according to the sequence similarity rather than to the availability of clones.

---

## [Author Response]

*Major points*:

*1) From the data presented, it is not clear how well a ligation event for an oligo pair will linearly report on a particular exon-exon joint when the pair is part of a complex set of ligamers. This has several complications*.

*A) From*
Figure 2*, it appears that increasing the total number of ligation events decreases the efficiency of product generation. This will lead to a favoring of exon skipping events in the assay. In*
Figure 3*, the authors compare the exon inclusion ratios of* CD45 *measured by RT/PCR and SeqZip and find that they are similar. However, RT/PCR is known to be biased towards exon-skipped products, especially when comparing size differences as large as those here. Do the similar results from the two assays indicate a similar exon-skipping bias from SeqZip*?

Yes, we agree that under suboptimal conditions (e.g., insufficient extension times) PCR can favor shorter over longer products. In the RT-PCR reactions shown in Figure 3, products representing the four *CD45* isoforms ranged from 367 to 850 nt. Our PCR conditions included a 1 min extension step, which should be more than sufficient to allow complete replication of all four isoforms (Taq DNA polymerase extension guidelines are typically 1 min per 1,000 nt). Supporting this, we observed no differences in isoform ratios after 18, 21, and 24 cycles of PCR (Figure 8). For the data shown in Figure 3, we used 21 cycles. Therefore, under the conditions we used here, there is little or no bias PCR toward shorter isoforms. Similarly, we observed no bias toward shorter isoforms with different PCR cycles in the SeqZip samples. Because SeqZip yielded the same isoform ratios as RT-PCR, we conclude that there was no bias toward shorter isoforms in the ligation reactions shown in Figure 3.

Author response image 1.**DOI:**
http://dx.doi.org/10.7554/eLife.03700.023

*B) Another source of error in the assay will likely be differences in annealing and ligation efficiency between different ligamer pairs. One oligo can have quite different ligation activities when joined to different partners. Different oligos have such drastically different annealing and ligation efficiency especially since they can also form various inter- and intra-molecular interactions when the complexity is increased. As a result, the poorest one will be the rate-limiting factor for the whole isoform presentation. For individual isoform pairs under different conditions, we use ratio/ratio to infer changes, but in the current study, this is not feasible. They may argue that a poor oligo will affect all isoforms equally, and thus, it is not a problem. However, other isoforms that lack the targeting exon for such poor oligo will be over represented. This problem may not surface with low complexity system. But, it is likely that it will soon be encountered by others, if the system is adopted, and if not properly addressed, the method will have little impact. This is a critical point for the authors to assess*.

To maintain modular cassette-exon ligamer design, ligation sites *must* be at splice sites. Also, analysis of exonic splice-site sequences support more uniform terminal exonic nucleotides, especially for the 5´ splice site (Lim & Burge, 2001). However, results should be cross-compared to traditional isoform analysis methods (RT-PCR). Finally, problematic “poor” oligo hybridization could be addressed using modified nucleotides such as LNA.

*C) Related to B, they make passing reference to matching the free energy of binding for each hybridizing segment, but it is not clear how well this works or whether it makes all oligo pairs join equally well. Similarly, the authors discuss the question of distinguishing highly similar exon sequences in DSCAM, but it is not clear if their approach is a general solution. Through assays of constructed pools they show they can distinguish exons 4.1 and 4.7, exons 6.9, 6.33, and 6.24, and exons 9.6 and 9.9, using their complex set of ligamers. However, it is not clear if these are the most similar of all DSCAM exon pairs or that others of the many possibilities might be more difficult to distinguish*.

We have supplied a multiple sequence alignment and sequence logo for the variant exons in clusters 4, 6, and 9 as Figure 9. These data show that there is low sequence variability between the alternative exons within a cluster. However, the high-degree of similarity between our data and those presented by Sun2013 support our *Dscam1* ligamer design's ability to discriminate between these highly-similar exons (53). The exons in our constructed pools were chosen based on availability of clones and overall high expression in S2 cells, not by sequence similarity.

Author response image 2.**DOI:**
http://dx.doi.org/10.7554/eLife.03700.024

*2) Although the authors present a complex oligo set for DSCAM whose ligated product can be assayed in a length accessible by MiSeq, it is not clear how general these conditions are for other complex systems, where there are more sites of variation and more types of splicing change. It seems likely that some regions will still need to be assayed separately and combinations predicted statistically. What are the limitations here? How complex a set of ligamers is feasible? Is there an upper limit on length that can be sequenced as a single fragment? Given such a length limit, what determines the number and spacing of alternative events that can be assayed by a ligamer set targeting all the possible products? Exactly how did the DSCAM mapping and variant quantification work? It appears that DSCAM isoforms were identified and counted by aligning to a library of isoform sequences and somehow scoring for sequence match. This needs to be described in much more detail. For this to be a useful method, it will need to be applicable to other complex transcript sets. It is not clear how feasible this is. Without understanding the mapping strategy, it is not clear how one might develop this for another gene*.

We have now added a paragraph to the Discussion specifically addressing these points.

Minor points:

*1) The paper is very haphazardly written and it is often difficult to follow exactly what is being done. At the same time, the authors often overstate the novelty and relative performance of the assay. Oligo Ligation as an assay for specific sequences in a complex mixture goes back at least to Landegren and Hood. The method here is basically an extension of RASL-seq. It doesn't take away from the advance to reference these precedents*.

We have added references to previous ligation-based assays in both the Results and Discussion sections.

*The authors comment on the better performance on SeqZip over CamSeq and PacBio long read sequencing. However, examining*
Figure 2
*and Supplement 6A, they appear comparable*.

We assume the reviewers intended to write ‘3E’ as Figure 2 does not have a panel E. Yes we fully agree that SeqZip, CamSeq, and PacBio all yield comparable usage for individual exons. But the clear advantage of SeqZip is that it is much less prone to template switching during library preparation and amplification.

*When discussing CamSeq, in the section “SeqZip eliminates* Dscam1 *isoform template switching analysis artifacts”, it is not clear what the authors mean by the sentence: “Although template-switched and unexplained isoforms together represented just 0.6-0.94% of all reads, they account for 99.9% of multi-exon isoforms sequenced…” The authors state that detection of template switched isoforms is much lower for SeqZip, but later report that 1.3% of the alignments in the sequencing of the DSCAM ligation products contained exons not present in the input. From this it would seem that the error rate for SeqZip is potentially higher than CamSeq*.

There is a major difference between these two ‘error rates’. Using our differential barcode approach, which uses sets of ligamers with different barcodes but whose ligation products are amplified together, we measure template-switching based on improper barcode combinations. We conclude that the vast majority of the reads aligning to ‘unintended’ isoforms originate from incorporation of near-cognate ligamers. Therefore, while a slightly higher overall percentage of SeqZip reads align to non-input isoforms, the source of most of these alignments (i.e. near-cognate ligamer usage) is known. In contrast, we assigned CamSeq reads to ‘template switched’ isoforms based logical rearrangement of high abundance control isoform exons. As these two ‘error rates’ are from different sources (template-switching vs. near-cognate ligamer usage), they cannot be compared. Finally, we believe that modest changes in ligamer design (i.e. longer hybridization regions, LNA incorporation) would further reduce near-cognate ligamer incorporation. In contrast, approaches to reduce of template-switching beyond the low rate that CamSeq has already achieved are not obvious to us.

*2) It is often unclear what the figures are presenting. More labels and lane numbers are needed. In*
Figure 2*, it was not clear that each bar represented a different variant for each region and should thus sum to 100%. In this Figure, the EDA bars for the EDA*^*+/-*^
*cells show only one isoform (they are not labeled as to which is which). Shouldn't both isoforms be present in these cells*?

In Figure 3, we have now added the requested labels for isoform variants. We also now indicate which sets of bars should sum to 100%. Regarding the EDA^+/–^ bars in the EDA^+/–^ cells, like the Reviewer we also expected to observe both isoforms. Therefore we were surprised when we only detected EDA^-^ isoform expression. We verified the presence of the EDA^+^ and EDA- alleles in gDNA by PCR. We tentatively conclude that the EDA^+^ allele is transcriptionally silenced in these cells. We now point out this apparent discrepancy in the text.

*The point of*
Figure 4
*is also not clear*.

The point of this figure is to show the extent of sequence identity when using different methods of RNA->DNA conversion for *Dscam1* isoforms. SeqZip greatly reduces the length of completely identical sequences thus minimizing the likelihood of template switching. We have now explained this figure in more detail in the text.

*3) As mentioned above, how were the isoform abundances calculated? This should be described in detail with a figure showing the conversion from read counts to isoforms*.

We now provide a new supplementary figure (Figure 4—figure supplement 1) showing our workflow for converting from read counts to isoforms.

*Also needed are the data tables and sequencing statistics for the MiSeq runs. Exactly how were these sequences mapped to transcripts to identify the exon combinations*?

We apologize for not including these tables in the full submission. They are now included in the revised submission. Mapping of sequence to transcripts is explained in Figure 4—figure supplement 1.

*The authors report the use of DESeq to determine differential expression of DSCAM isoforms. They should be aware that this is considered to be a bad idea generally and specifically that SeqZip libraries likely violate several crucial assumptions made by DESeq. This should be clarified in detail*.

Our ability to assign each ligation product uniquely to a particular combination ensures the independence of the read counts between the isoforms, hence fulfilling a key assumption of DESeq. The isoform abundances vary over 5-logs, and exhibit similar dispersion to RNA-Seq samples. We therefore believe DESeq is a suitable tool for identifying population outliers, which in our case would be indicative of isoforms displaying linked exon usage.

*4) The authors stress that PCR amplification must be minimized, but report 12 cycles of PCR after ligation, followed by 22 cycles to add the Illumina primers. How were these numbers arrived at? Was it the same for the biological samples and the artificial pools*?

SeqZip of *Dscam1* by HTS is equivalent to single-gene RNA-Seq. That is, rather than making a library of the entire transcriptome, we made libraries from transcripts from a single gene expressed at a low level. One cannot get the amount of material needed without many cycles of PCR. We have now removed the phrase “and subsequent PCR amplification is kept to a minimum” from the text. We have also inserted text in the Methods explaining how we normalized between samples the amount of *Dscam1* DNA from the first PCR reaction into the second PCR reaction. We note that all of our ligation reactions were performed in duplicate with two different barcodes. The duplicate samples were mixed immediately upon completion of the ligation reaction. Having two barcodes in the same PCR sample enabled us to detect template-switched isoforms in the final library. If any library contained >0.1% template-switched isoforms it was considered over amplified and not used.

*How will this be affected by transcript abundance for targets that are less abundant than DSCAM? The DSCAM splice variants are all the same length, but this is not usually the case for systems of complex splicing. How will this PCR step affect the quantification when some variants are much shorter than others*?

*Dscam1* is a low to moderate abundance transcript in embryos (See Figure 10). Our ability to characterize specific *Dscam1* isoforms extracted from whole animals demonstrates the sensitivity of our method. Some amount of empirical investigation will likely be required to adapt SeqZip for extremely rare transcripts, but this would be true of any technique.

Author response image 3.*Dscam1* expression as measured by ENCODE high-throughput sequencing. Accessed 2014-10-12 via http://flybase.org/reports/FBgn0033159.html**DOI:**
http://dx.doi.org/10.7554/eLife.03700.025

SeqZip compresses the connectivity information in a long RNAs into ligation products <500 nt long but >100 nt. In this size range, preferential PCR of smaller fragments is much less of issue than for methods where full length cDNAs are measured and lengths might vary by thousands of nucleotides.

[Editors' note: further revisions were requested prior to acceptance, as described below.]

*The manuscript has been improved, but all the reviewers feel that there are some comments that would require a minor revision. Once these are introduced to the description of the results, methods and/or discussion (no further experiments or analyses are requested) we will be able to accept the paper. We emphasize that we do not anticipate re-sending the manuscript to the reviewers*.

*Specifically*:

*1) The authors should add a discussion of areas for future analytics to better improve the reliability and quantifiability of the inferred isoforms. These should accompany the currently well-described experimental limitations of SeqZip*.

As detailed below, we have now added the requested verbiage to the Discussion.

As Reviewer 1 notes in further detail (and agreed by all reviewers): (1) there may be genes and isoforms that cannot readily be assayed with ligamer pools that uniquely quantify combinations of splicing events, but proper inference would still be able to help estimate splicing frequency;

We have now made this clear in the Discussion.

Guidelines for a method/algorithm to design a valid ligamer pool for a given gene are important to highlight in terms of what is already available and what is a future analytic direction.

To help the reader understand ligamer pool design, we have now added a schematic (Figure 1figure supplement 7) showing our design process for terminal and internal mouse Fn ligamers. Because each individual application will necessitate a ligamer design specific to that application, we have not attempted to generate a universal computerized ligamer design implementation. Instead we think it more useful for the reader to fully understand how such an algorithm might work and then generate their own.

*2) Several critical details are missing or quite unclear, especially in the Methods sections, which were missing before or even removed in this version, per the items detailed in Reviewer 2's review. In particular, we believe that the information on number of PCR cycles is* absolutely critical *to reinstate to the Methods section*.

As noted below, the number of PCR cycles was previously included in the Methods section. But perhaps the reviewer could not locate them because the order of the Methods section did not parallel the order of the Results section. This has now been corrected. We have also expanded the Methods section to include detailed descriptions of the experiments presented in all figures.

*3) There are some lingering concerns with the extent to which the method is (1) quantitative and (2) has false negatives (the two are inter-related). Again, we would like to see the discussion section clarify to the reader, as much as possible, what they can and cannot expect from the method. This is important for its long-term impact in the field*.

We have now added the following verbiage to the Discussion to address this point:

“In our experiments, we were able to demonstrate accurate isoform abundance reporting over >4 orders of magnitude for *Dscam1* (Figure 5). At the low end of this range, however, near-cognate ligation events began to be problematic. As for sensitivity, we have been able to obtain detectable ligation products from as few as ∼900 (5 × 10^−17^ M; 0.05 fM) target RNA molecules (data not shown). Because the limit of detection for SeqZip is likely more than a single molecule, lack of detection of a particular isoform should only be interpreted as that isoform being below the SeqZip detection limit. Nonetheless, when properly controlled, SeqZip is a sensitive quantitative method for assessing complex isoform abundance patterns over a wide dynamic range.”

Reviewer 1:

The authors have clarified a number of concerns I had with this revised version of the manuscript. The authors' section on the limitations of SeqZip clearly specifies most of the experimental limitations that remain. I would have liked to see a more thorough discussion of the analytic limitations or areas for improvement in the Discussion. For example:

*1) As the authors note, there may be genes and isoforms that cannot readily be assayed with ligamer pools that uniquely quantify combinations of splicing events. However, SeqZip could still provide major advantages over conventional RNA-Seq for these genes, if it were coupled with statistical inference to estimate the frequency of splicing event combinations. The requirement that reads uniquely quantify isoforms is helpful but probably unnecessary for recovering accurate values. Using DESeq will only work well with ligamer pools that uniquely specify splicing combinations. For more complex pools, the authors’ current pipeline is likely to return unreliable numbers*.

As the reviewer notes, application of DESeq for analyzing DSCAM ligamer isoforms is unique to that experimental design. Further, how best to analyze a particular SeqZip experiment (e.g., radioactive PCR on polyacrylamide gels vs deep sequencing) will depend on the exact experimental design and the number of isoforms being queried.

*2) The authors do not describe an algorithm that generates a valid ligamer pool for a given gene. How are users of SeqZip to know that the pool they design meets the requirements of the assay and the analysis pipeline? What if they make a mistake, and leave a particular combination of splicing outcomes “uncovered” by the ligamer pool? A rigorous design tool for making ligamer pools would ensure that the current analysis pipeline returns valid quantification. Alternatively, the analysis pipeline itself could be improved (as noted in #1) to support more relaxed ligamer designs*.

Nowhere in this manuscript did we or do we now present any computational “pipeline” for designing ligamer pools or for analyzing the results. Because of the flexibility of the method for addressing many different kinds of questions, it is hard for us to envision (and therefore even more difficult to implement) any single “pipeline” that would be suitable for all applications. Nonetheless, we do see the utility of giving the reader a better idea of how we went about designing large ligamer pools. Therefore we now include as Figure 1–figure supplement 7 a detailed flow diagram of our ligamer design process for the mouse Fn gene.

*Both of these issues seem like good potential future directions. I would just suggest that the authors note them as limitations or opportunities in the Discussion*.

The following paragraph has now been added to the Discussion:

“The easiest ‘complex’ form of alternative splicing for SeqZip analysis was *Dscam1*, where alternative processing is limited to mutually exclusive exons […] For highly complex pools, this process can be automated by writing a simple Python, Perl, or R script specific to the problem being addressed. ”

Reviewer 2:

*While the authors addressed most of the concerns raised earlier, this “final” version still has various problems*:

*1) The optimal PCR cycle number. The authors argued that under their conditions, the PCR cycle up to 22 did not cause any problem, yet still recommended to keep the cycle minimal in the previous version. Now, the authors removed such recommendation because they no longer think it is a critical issue? They did not even specify in their method section how many PCR cycles used. This information is needed for others to use the method*.

In the previous manuscript version, the number of PCR cycles used was clearly stated in the relevant paragraph of the Methods section for both “MiSeq Library Preparation” and “Radioactive PCR”. In rereading the Methods section, we realized however, that the order in which specific methods were presented was not the order in which they appeared in the Results. This may have led to the reviewer's confusion as to where to find the PCR cycle numbers. We have now reordered the Methods section to make it parallel the Results section, and PCR cycle numbers are clearly stated for every experiment.

*2) Template switching was not a major problem with the amount of starting materials the authors used. However, most experiments use total RNA from a large number of cells. The authors need to provide instruction in their Methods section for the optimal range of RNA concentrations to be used in their assays*.

The range of total RNA starting concentrations is now explicitly stated in the “SeqZip” methods paragraph.

*3) I am still confused by the phrase in the section “SeqZip eliminates template-switching artifacts in the analysis of* Dscam1 *isoforms”: “Although the percentage of total reads aligning to template-switched and unexplained isoforms was low (0.6-0.94%), these few reads support the majority of* Dscam1 *isoforms sequenced (99.9%)”. Did the authors mean the majority of artifacts sequenced? If this is what they meant, SeqZip detected 1.3% artifacts, which is higher than CAMSeq*.

No we meant that 99.9% of species detected in the CAMSeq control experiments were artifacts. Because this passage was the cause of much confusion, we have now eliminated it and reworked the paragraph to make it clearer.

*4) In the Discussion, the authors cited “slow RNA synthesis favors inclusion of cassette exons with weak splicing signals, while fast RNA synthesis favors exclusion (Kornblihtt et al. 2013)”. They need to cite a recent paper in Mol Cell from the lab that has corrected this simple version of the model*.

Done.

*5) The authors added additional references in the text, but many did not show up in the reference list. Seem to need to update their EndNote. They should cite the Hood Science paper, which is the first to describe the ligation approach*.

Fixed.

Reviewer 3:

*The SeqZip method presented by the authors does address an important issue that is of general interest. The concept for the method has been around previously. Thus, I consider it essential that a high-profile publication reporting this method contains extensive validation of the approach*.

*The authors have made an attempt to address the issues raised in the review. However, the major issue remains: under which circumstances can the SeqZip method be considered as quantitative? To ensure the quantification aspect of SeqZip, several conditions have to be examined*:

*i) the specificity of the ligamers*;

*ii) the equal hybridization and ligation efficiencies according to ligamer sequences*;

*iii) the equal ligation efficiencies across various numbers of ligation events*.

*These key parameters differ for each target and rigorous control criteria are essential to set up suitable conditions for each target. Without prior validation of these criteria, SeqZip cannot be considered as quantifiable. The only conclusion that could thus be addressed is that combination of alternative events is present within one transcript if there is a signal (but absence of signal doesn't necessary mean that the isoform doesn't exist)*.

*The authors have selected different targets to validate different aspects of the method (*CD45*,* FN1 *and* Dscam1*) but their major figure describing SeqZip method (*Figure 2*) highlights the strong bias to underrepresent mRNA isoforms that require multiple ligation events*.

We assume that by “Figure 2”, the author means Figure 2. In this experiment, each ligamer contained only 10 nts of complementarity to either side of the looped out region, there was no amplification of ligation products, and all ligamers were the same length. Therefore all three 2-way and both 3-way ligation products co-migrate on the gel, whereas there is only one possible 4-way ligation product. This co-migration likely explains the reviewer's conclusion that there is a “strong bias to underrepresent mRNA isoforms that require multiple ligation events”. The results in Figure 3, however, clearly demonstrate that when full-length ligation products were amplified, there was no detectable bias against longer isoforms in favor of shorter isoforms. Indeed the number of ligation events required for the longest *CD45* isoform (R456) is 3 more than the number required for the shortest (R0). Of course, the safest comparisons will always be between ligation reactions requiring similar numbers of ligation events (e.g., *Dscam*), and care should be taken when comparing abundances of ligation products resulting from significantly different (Δ>3) ligation events. We have now added the following cautionary verbiage in the Discussion:

“One limitation of SeqZip is the number of ligamers required to create a ligation product […], it is crucial to perform the necessary controls to ensure quantitative detection of all desired isoforms.”

*The Figure 9 increases the concern that also ligamers can be non-specific. Nevertheless, the authors strongly argue against relying on the CAMseq experiments, which at the same time they consider as not reliable. Again, a more rigorous, systematic inclusion of controls is needed. For example, the authors should have selected control constructs according to the sequence similarity rather than to the availability of clones*.

We agree that other control constructs might have been more optimal, but at the time of these experiments such constructs were not available to us. We disagree with the reviewer, however, that we “strongly argue against relying on the CAMseq” approach. It was our intent to provide a disinterested analysis of the advantages and disadvantages of RT-PCR-based approaches (i.e., triple read sequencing and CAMseq) versus a RNA-templated ligation-based approach (i.e., SeqZip). While SeqZip solves the template-switching problem inherent to the RT-PCR-based methods, it obviously has its own limitations. These are now covered in great detail in the “SeqZip uses and limitations” section of the Discussion. We leave it to the reader to decide which approach is most appropriate for his or her needs.